# Evolutionary druggability for low-dimensional fitness landscapes toward new metrics for antimicrobial applications

**Rafael F Guerrero[1], Tandin Dorji[2], Ra'Mal M Harris[3], Matthew D Shoulders[3], C Brandon Ogbunugafor[3,4,5,6]\***

[1]Department of Biological Sciences, North Carolina State University, Raleigh, United States; [2]Department of Mathematics and Statistics, University of Vermont, Burlington, United States; [3]Department of Chemistry, Massachusetts Institute of Technology, Cambridge, United States; [4]Department of Ecology and Evolutionary Biology, Yale University, New Haven, United States; [5]Santa Fe Institute, Santa Fe, United States; [6]Public Health Modeling Unit, Yale School of Public Health, New Haven, United States

**Abstract** The term 'druggability' describes the molecular properties of drugs or targets in pharmacological interventions and is commonly used in work involving drug development for clinical applications. There are no current analogues for this notion that quantify the drug-target interaction with respect to a given target variant's sensitivity across a breadth of drugs in a panel, or a given drug's range of effectiveness across alleles of a target protein. Using data from low-dimensional empirical fitness landscapes composed of 16 β-lactamase alleles and 7 β-lactam drugs, we introduce two metrics that capture (i) the average susceptibility of an allelic variant of a drug target to any available drug in a given panel ('*variant vulnerability*'), and (ii) the average applicability of a drug (or mixture) across allelic variants of a drug target ('*drug applicability*'). Finally, we (iii) disentangle the quality and magnitude of interactions between loci in the drug target and the seven drug environments in terms of their mutation by mutation by environment (G x G x E) interactions, offering mechanistic insight into the variant variability and drug applicability metrics. Summarizing, we propose that our framework can be applied to other datasets and pathogen-drug systems to understand which pathogen variants in a clinical setting are the most concerning (low variant vulnerability), and which drugs in a panel are most likely to be effective in an infection defined by standing genetic variation in the pathogen drug target (high drug applicability).

## eLife assessment

This manuscript introduces two **valuable** new metrics - "variant vulnerability" and "drug applicability" - that would be of use to identify candidate drugs for treating infections while considering longer-term, evolution-based treatment outcomes. Despite the intuitive appeal of the metrics and their potential, the study remains **incomplete**, as it fails to demonstrate the generality of the approach. The work could be improved by analysing a broader range of data in a systematic way and directly tying the metrics to outcomes, which would make it possible to better assess their impact and utility.

## Introduction

Evolutionary concepts have long been applied to the problem of antimicrobial resistance. For example, our understanding of how resistance evolves has improved through studies of protein

**\*For correspondence:** brandon.ogbunu@yale.edu

**Preprint posted** 08 April 2023
**Sent for Review** 25 April 2023
**Reviewed preprint posted** 14 August 2023
**Reviewed preprint revised** 30 November 2023
**Version of Record published** 04 June 2024

evolution (*Weinreich et al., 2006*; *Lozovsky et al., 2009*; *Toprak et al., 2011*), reversion (*Tan et al., 2011*; *Baym et al., 2016*; *Ogbunugafor and Hartl, 2016a*), and higher order interactions between drugs (*Michel et al., 2008*; *Yeh et al., 2009*) or mutations (*Weinreich et al., 2013*; *Lozovsky et al., 2021*). While stepwise, de novo evolution (via mutations and subsequent selection) is a critical force in the evolution of antimicrobial resistance, evolution in natural settings often involves other processes, including horizontal gene transfer (*Bergstrom et al., 2000*; *Turner et al., 2014*; *Loftie-Eaton et al., 2017*) and selection on standing genetic variation (*Pennings, 2012*; *Pelleau et al., 2015*). Consequently, perspectives that consider variation in pathogens (and their drug targets) are important for understanding treatment at the bedside. Recent studies have made important strides in this arena. Some have utilized large data sets and population genetics theory to measure cross-resistance and collateral sensitivity (*Gjini and Wood, 2021*; *Ardell and Kryazhimskiy, 2021*; *Herencias et al., 2021*; *Nichol et al., 2019*). Fewer studies have used evolutionary concepts to establish metrics that apply to the general problem of antimicrobial treatment on standing genetic variation in pathogen populations, or for evaluating the utility of certain drugs' ability to treat the underlying genetic diversity of pathogens (*Yeh et al., 2006*; *Yeh et al., 2009*; *Mira et al., 2021*).

Despite these gaps in the world of antimicrobial resistance, a robust literature in immunobiology—specifically concerning the problem of evaluating broadly neutralizing antibodies to counteract the widespread genetic diversity of viral infections—has addressed analogous concepts. Studies have profiled mutations in influenza that allow them to escape antibodies (*Doud et al., 2017*; *Doud et al., 2018*), and reconstructed 'binding affinity landscapes' that define the constraints around viral escape (*Phillips et al., 2021*; *Phillips et al., 2023*). These studies, which focus on both the antiviral antibody and their targets (viruses), demonstrate that metrics capturing the susceptibility of a target, or efficacy of an antimicrobial actor, are highly relevant in our quest to prevent, diagnose, and treat infectious diseases.

This study examines data from protein fitness landscapes across a panel of antimicrobial drugs to develop new metrics for antimicrobial applications. Specifically, we propose novel druggability framings, a concept that relates to the interaction between drugs and their targets. We propose an extension of this concept in the form of two metrics: (i) The average susceptibility of alleles of a drug target across drugs in a panel ('*variant vulnerability*') and (ii) the efficacy of drugs across alleles of a given drug target ('*drug applicability*'). Lastly, we statistically deconstruct the peculiar interactions that underlie these metrics: interactions between drugs and the genetic variants at their target ($\beta$-lactamases in this case). We label this last analysis as measuring the 'environmental epistasis' in the drug-target interaction, a concept developed recently to describe molecular evolution in the presence of environmental change (*Lindsey et al., 2013*).

Summarizing, we propose that these new metrics can be applied to pathogen-drug datasets to identify challenges associated with specific variants of pathogens, and evaluate the efficacy of certain drug types. Further, our results show how modern concepts in evolutionary genetics, like gene by gene by environment interactions (environmental epistasis) have practical utility in biomedicine.

## Results

Here, we describe several analyses, emphasizing two novel metrics we have developed: variant vulnerability and drug applicability. We first plotted the growth rates for the 16 alleles in the TEM-1/TEM-50 fitness landscapes across the seven drug environments (*Figure 1*). We then computed the variant vulnerability and drug applicability metrics and deconstructed the metrics by measuring the interactions between mutations and drug environments (including G x G x E interactions).

### Variant vulnerability

Low variant vulnerability implies low susceptibility of a genotype to the library of drugs in the panel (*Figure 2*). Of the alleles in our set, 0110 (MKSN) displayed the lowest susceptibility with resistance to 3/7 drugs—ceftazidime, cefotetan, and cefotaxime. Alternatively, the allele with the highest variant vulnerability was a triple mutant, 0111, corresponding to the MKSD allele.

We can rank alleles from highest to lowest for a different interpretation of the variant vulnerability values (*Table 1*). Notably, the allele that had the highest variant vulnerability (0111, MKSD), and the one with the lowest variant vulnerability (0110, MKSN) differ by a single mutation (N276D). That a

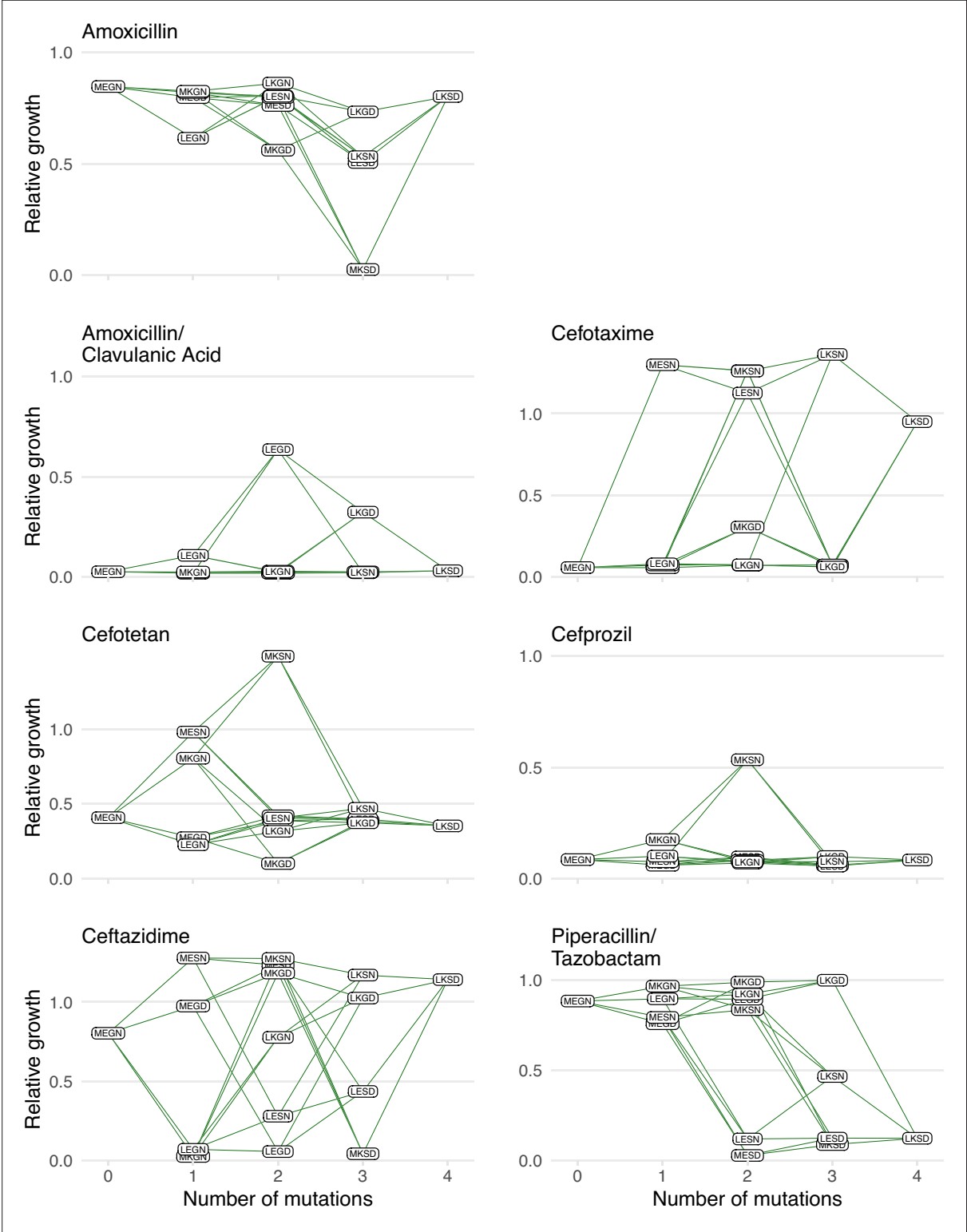

**Figure 1.** Graphical depictions of the TEM1/TEM-50 fitness landscape examined in this study. The TEM-50 variant is represented by the quadruple mutant, LKSD, and the wild-type, MEGN, represents TEM-1. The figure highlights the variation in the topography of the various fitness landscapes. Y-axes correspond to the growth rates relative to the wild-type of the allelic variants. The X-axis corresponds to the number of mutations away from the TEM-1 wild-type MEGN.

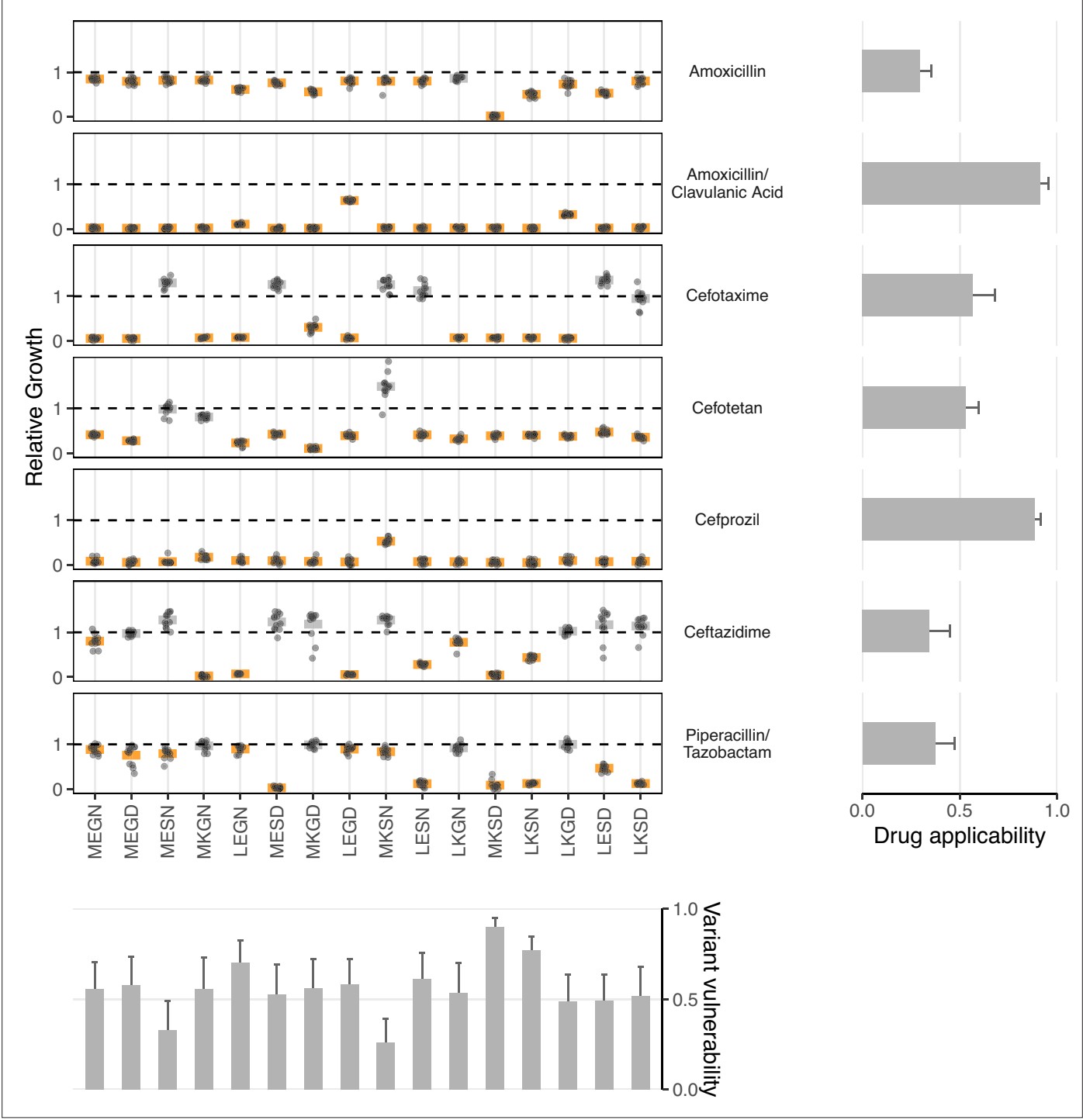

**Figure 2.** Data from empirical fitness landscapes can be used to compute two metrics: variant vulnerability (for allelic variants) and drug applicability (for drugs). Growth of allelic variants, relative to control (wild-type), for seven drug treatments. Bars represent the mean value for each group; orange bars indicate significantly lower growth with respect to control (one-tailed t-test, p<0.01), grey bars otherwise. The bars, from left to right, along the bottom correspond to the variant vulnerability of the variants. The bars along the right side, from top to bottom, describe the drug applicability of the seven drugs analyzed in this study. Dashed lines correspond to the TEM-1 growth rate in the putative drug environment.

**Table 1.** The rank order of the alleles with respect to variant vulnerability.

An allele with high drug applicability is most susceptible to the collection of drugs in a panel (seven $\beta$-lactamase drugs and drug combinations in this study). MEGN (0000) is the genotype known as the TEM-1 variant of $\beta$-lactamase. LKSD (1111) is the TEM-50 variant of $\beta$-lactamase genotype.

| TEM allelic variant | Binary | Rank (1=highest variant vulnerability) |
| --- | --- | --- |
| MKSD | 0111 | 1 |
| LESD | 1011 | 2 |
| LEGN | 1000 | 3 |
| MEGD | 0001 | 4 |
| MKGN | 0100 | 5 |
| LESN | 1010 | 6 |
| LKGN | 1100 | 7 |
| LEGD | 1001 | 8 |
| MKGD | 0101 | 9 |
| LKGD | 1101 | 10 |
| LKSD | 1111 | 11 |
| MEGN | 0000 | 12 |
| MESD | 0011 | 13 |
| LKSN | 1110 | 14 |
| MESN | 0010 | 15 |
| MKSN | 0110 | 16 |

Source: The authors.

single mutation can significantly impact variant vulnerability highlights how single mutations dramatically affect the topography of the variant vulnerability fitness landscape, with the N276D mutation negatively affecting the susceptibility through interactions with the genetic background. Also, note that the alleles associated with TEM-1 (MEGN) and TEM-50 (LKSD) have variant vulnerability values that are close (ranked 12th and 11th respectively; among the lowest values).

How do we interpret the low variant vulnerabilities of TEM-1 and TEM-50—indicative of low susceptibility to the panel of drugs? Whatever the selective pressures that drove the evolution of TEM-1 and TEM-50 in natural settings, these findings suggest that the resulting resistance phenotypes associated with evolved $\beta$-lactamases seem to confer general survivability across drugs. This observation is consistent with prior notions of 'cross resistance' (*Oz et al., 2014*; *Cândido et al., 2019*; *Colclough et al., 2019*).

### One-step neighbor variant vulnerability

Our findings regarding large variant vulnerability differences between mutational neighbors like MKSN (0110; the lowest variant vulnerability value of the 16 alleles) and MKSD (0111; the highest variant vulnerability value of the 16 alleles) inspired further analysis of the relationship between the variant vulnerability of an allelic variant and its mutant neighbors. As the mutants analyzed in this study compose a combinatorially complete set, they can be depicted in the formal structure of a canonical fitness graph that outlines changes between TEM-1 and TEM-50 (*Figure 1*). And while this study focuses on the individual properties of variants/alleles of TEM-1/TEM-50, one might ask how the variant vulnerability values are distributed across the fitness graph, which may speak to its evolvability. Is there a correlation between the variant vulnerability values and those of one-step mutant neighbors? Alternatively, are variant vulnerability values uncorrelated from that of their neighbors?

To answer these questions, we calculated each allele's average variant vulnerability of one-step mutational neighbors. *Figure 3* depicts the variant vulnerability values for the 16 allelic values (from

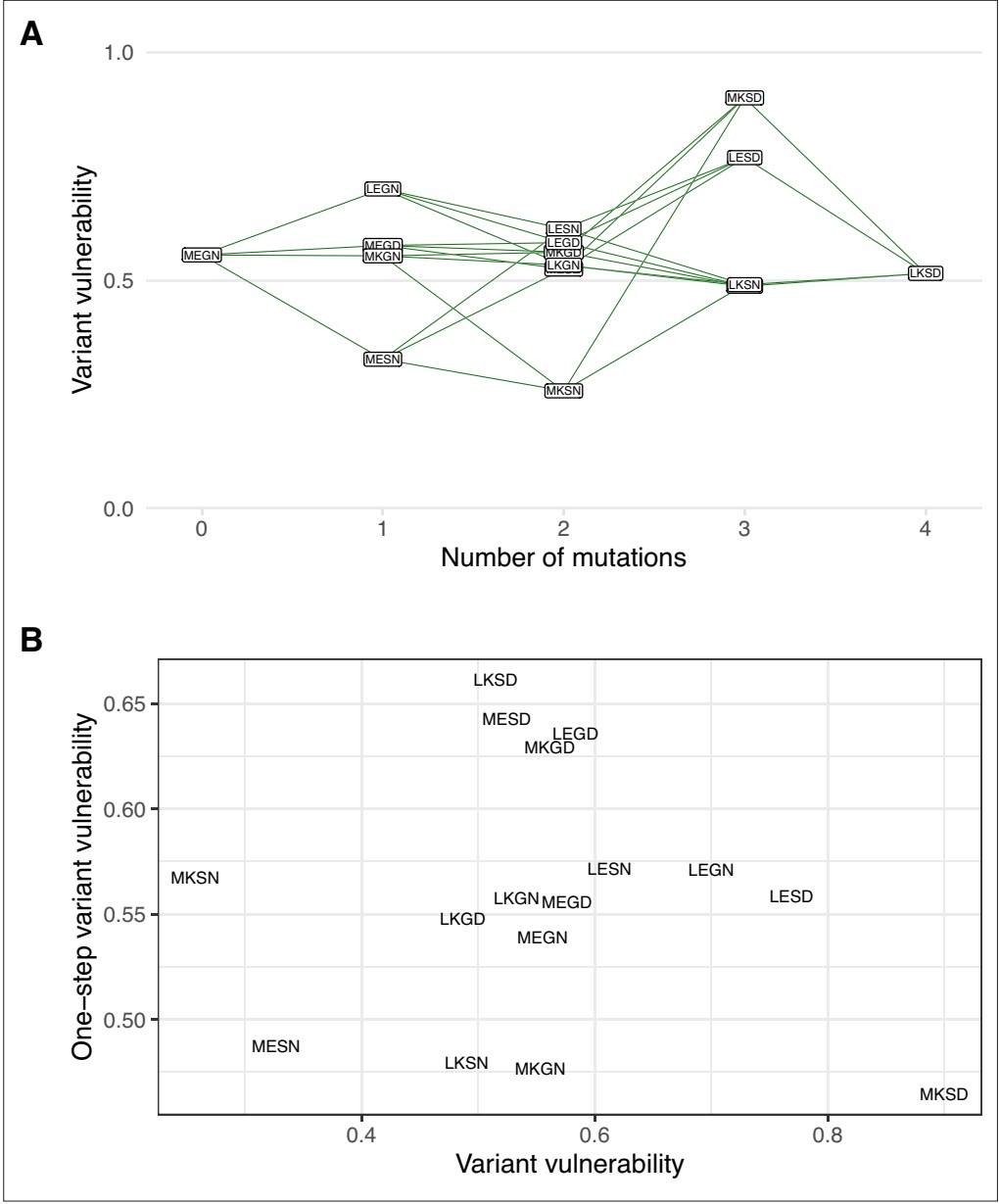

**Figure 3.** The variant vulnerability of the 16 allelic variants does not correlate with that of their one-step mutational neighbors. (**A**) The variant vulnerability values for the 16-allelic variants as a function of the number of mutations away (hamming distance) from TEM-1 (MEGN). (**B**) The variant vulnerability values of all 16 allelic variants (x-axis) against the average variant vulnerability values of one-step neighbors (each allelic variant has four such neighbors). The analysis demonstrates no correlation, suggesting that variant vulnerability values are distributed nonlinearly across the fitness landscape (linear regression $R^2 = 0.01$, $P = 0.69$).

*Figure 2*), and a scatterplot depicting the relationship between allelic variants and the average variant vulnerability of one-step neighbors (each variant has four neighbors). Correlation between variant vulnerability values and those of their nearest neighbors (linear regression $R^2 = 0.01$, $P = 0.69$). The lack of correlation suggests that the fitness landscape for variant vulnerability is rugged, where individual mutations contribute nonlinearly to the susceptibility of a given allelic variant.

## Drug applicability

A high applicability value indicates that a drug is effective across the suite of genetic variants in a population (*Figure 2*). This study corresponds to a combinatorial set of 16 TEM-1/TEM-50 mutants.

**Table 2.** The rank order of the seven drugs with respect to their drug applicability.
A drug with a high drug applicability is, on average, highly effective across the range of allelic variants in a set (in this study, a combinatorial set of four mutations that compose the TEM-50 variant of $\beta$-lactamase in this study). We emphasize that the findings in this study do not apply to clinical settings nor are they based on clinical data.

| Antimicrobial | Class | Rank (1=highest drug applicability) |
|---|---|---|
| Amoxicillin / clavulanic acid | $\beta$-lactam and $\beta$-lactamase inhibitor | 1 |
| Cefprozil | Second-generation cephalosporin | 2 |
| Cefotaxime | Third-generation cephalosporin | 3 |
| Cefotetan | Second-generation cephalosporin | 4 |
| Piperacillin and tazobactam | Penicillin and $\beta$-lactamase inhibitor | 5 |
| Ceftazidime | Third-generation cephalosporin | 6 |
| Amoxicillin | $\beta$-lactam | 7 |

Source: The authors.

In *Table 2*, we rank-order the drugs with respect to their drug applicability. The treatment with the highest drug applicability was the amoxicillin/clavulanic acid combination, comprising both a $\beta$-lactam (amoxicillin) and a $\beta$-lactamase inhibitor (clavulanic acid). This combination affected the growth rate of all 16 allelic variants. The treatment with the lowest drug applicability was amoxicillin alone. This observation fits intuition, as the combination of amoxicillin and clavulanic acid contains both a $\beta$-lactam and a $\beta$-lactamase inhibitor.

### Analysis of environmental epistasis

Finally, we conducted a statistical decomposition of the individual effect sizes associated with individual SNPs, SNP x SNP interactions (epistasis), SNP x environment (plasticity), and SNP x SNP x environment (environmental epistasis) interactions (*Figure 4*). Here, we observe that the four-way interaction between all four loci has a powerful positive effect on recovering growth rate defects in three drugs—amoxicillin, cefotaxime, and ceftazidime (5.8, 8.4, and 5.5 standard deviations respectively). Note how the four-way interaction only has a small effect on the drug mixtures that include $\beta$-lactamase inhibitors: amoxicillin/clavulanic acid and piperacillin/tazobactam. For amoxicillin/clavulanic acid—the drug regime with the highest drug applicability, only two mutation interactions have a meaningful impact on the effect of those drugs across variants: a pairwise interaction between sites M69L and N276D (labeled A x D in *Figure 4*), and a three-way interaction between M69L, E104K, and N276D (A x B x D). Most interactions either have no effects or negative effects. This observation is consistent with the amoxicillin/clavulanic drug having the highest drug applicability, with few mutation interactions meaningfully providing resistance.

## Discussion

In this study, we examine a data set from an empirical fitness landscape of an antimicrobial drug target to develop metrics for identifying (i) which allelic variants are most likely to be resistant to whole panels of drugs, and (ii) which drugs are most likely to be effective against a suite of resistance allelic variants of a given drug target. We then measure the interactions between individual loci and drug environments that underlie these two metrics.

We focused on mutations associated with the TEM-1/TEM-50 class of $\beta$-lactamases, associated with resistance in bacteria, and seven $\beta$-lactam drugs. We study a combinatorial set of four different mutations, allowing us to compute the epistasis x environment (also known as 'environmental epistasis'; see *Lindsey et al., 2013*) relationships between alleles.

With respect to variant vulnerability, which measures how susceptible a given allele is across drug types, we observe several intriguing phenomena. Firstly, the allelic variant with the highest variant

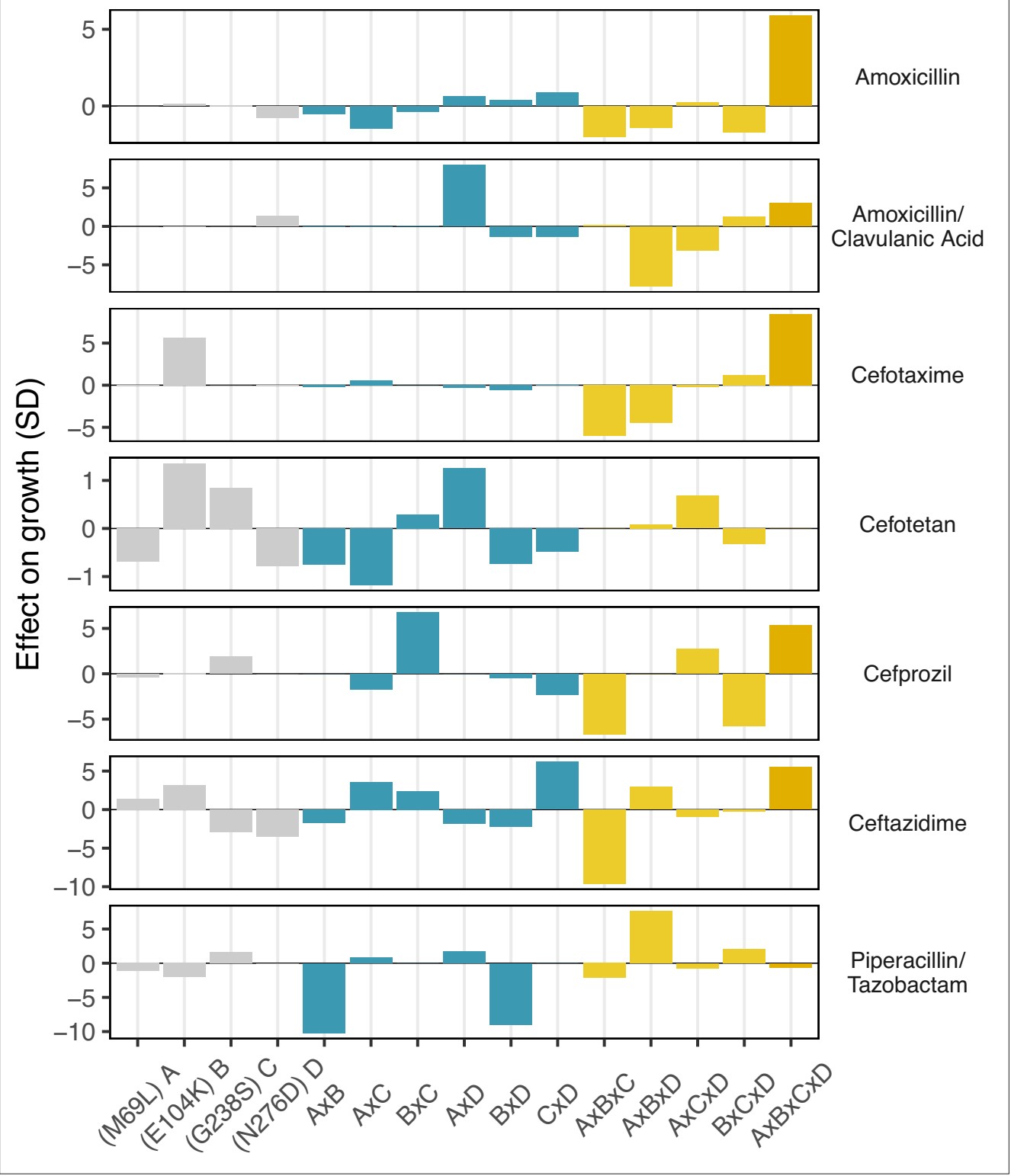

**Figure 4.** Environmental epistasis underlies the drug-allele interactions that drive the variant vulnerability and drug applicability. Effect on growth (in standard deviations of the wild-type control values), estimated by LASSO regression, for individual loci and their interactions. [A] corresponds to M69L, [B] to E104K, [C] to G238S, and [D] to N276D. As in a mutation effect reaction norm (*Ogbunugafor, 2022*), the information describes how the effect of mutations changes across drug environments. This analysis is akin to a large deconstruction of the SNP x SNP x Environment (G x G x E) interactions, also known as environmental epistasis (*Lindsey et al., 2013*). In the supplemental information, we provide a version of this figure where all of the coefficients are the same axis.

vulnerability (most susceptible across drug types; MKSD or 0111) and lowest variant vulnerability (MKSN or 0110) are only a single mutation apart (N276D) (*Figure 1*). That a single mutation can have large consequences for treatment across drugs in a panel is not surprising from one perspective—individual mutations can certainly have strong effects. The analysis of environmental epistasis (*Figure 4*) highlights, however, that the dramatic difference in phenotype between MKSD and MKSN is as much about how that mutation effect depends on the other amino acids in the protein background. The context-dependence of mutations contributing to variant vulnerability highlights the continued need to resolve the population genetic particulars of a given infection. For example, in a clinical setting, inaccurate identification of the SNP or amino acid at a single locus (N276D in our study) could create an entirely different picture of the clinical course of infection.

Our findings on drug applicability highlight the complexity underlying drug mechanisms and their consequences. Though all seven drugs/combinations are $\beta$-lactams, they have widely varying effects across the 16 alleles. Some of the results are intuitive: for example, the drug regime with the highest drug applicability of the set—amoxicillin/clavulanic acid—is a mixture of a widely used $\beta$-lactam (amoxicillin) and a $\beta$-lactamase inhibitor (clavulanic acid; see *Table 2*). We might expect such a mixture to have a broader effect across a diversity of variants. However, this high applicability is hardly a rule, as another mixture in the set, piperacillin/tazobactam, has much lower drug applicability (ranking 5th out of the seven drugs in the set; *Table 2*). Intuition can over-simplify, with undesirable consequences: not all drug combinations/mixutures are more effective over a larger breadth of alleles. Because this study focused on mutations in combination, rather than drugs, we did not examine the drug applicability of the individual drugs in all the drug combinations (e.g. piperacillin alone and tazobactam alone). However, there is a rich literature covering the specific problem of drug combinations (*Michel et al., 2008*; *Beppler et al., 2016*; *Singh and Yeh, 2017*; *Tekin et al., 2017*).

The surprising results in both the variant vulnerability (regarding the ability of allelic variants to grow across niche breadth) and drug applicability (regarding the impact of drugs across allelic variants) highlight the importance of specific interactions between individual loci (individual sites in TEM-50) and drug environment ($\beta$-lactam drugs). For example, we observe the strongly positive interaction between mutations at the four sites associated with TEM-50 in several drugs: amoxicillin, cefotaxime, cefprozil, and ceftazidime. Other higher-order interactions are associated with different effects: note the strongly negative effects of the three-way interaction between mutations at loci M69L, E104K, and G238S on ceftazidime growth rate (*Figure 4*). Regarding the main effects, we note the positive effect of the M69L mutation on the growth rate in cefotaxime, cefotetam, and ceftazidime. Alternatively, the G238S, N276D mutations have individual negative effects on ceftazidime. Interestingly, in combination, these mutations strongly positively affect ceftazidine (*Figure 3*). While our study does not delve into the biophysical underpinnings of any of these findings, they do support other studies that speak to the importance of how even slight structural differences between drugs can have strong implications for how they interact with allelic variants of a single protein (*Ogbunugafor et al., 2016b*; *Lozovsky et al., 2021*; *Mira et al., 2021*).

## Study limitations

Firstly, this study utilized data from a set of low-dimensional empirical fitness landscapes (16 alleles, across seven drugs/combinations). We emphasize that our aim is not to provide any insight about treatment regimes, or inform the real-world use of any specific antimicrobial. Therefore, any direct clinical applications of our findings are limited. Instead, our study offers an evolutionary perspective and metrics that are largely agnostic with respect to drug type and can be further applied to drug-pathogen data of various kinds. Given the rise of technologies (e.g. deep mutational scanning; *Fowler and Fields, 2014*) that permit the experimental generation of higher dimensional fitness landscapes, future studies should apply these metrics and analyses to larger data sets where we might determine the variant vulnerability of thousands of variants, alongside the drug applicability of hundreds of drugs. We hope that we have outlined our methods in a manner that makes such applications relatively easy to consider.

In addition, our study focused on a discrete set of individual drugs and mixtures. A robust literature exists surrounding the importance of drug-drug interactions in antimicrobial resistance (*Michel et al., 2008*; *Yeh et al., 2009*; *Beppler et al., 2016*; *Tekin et al., 2017*). Considering how these metrics apply in the context of higher order combinations of drugs constitutes an area of future inquiry.

Lastly, we use the $\beta$-lactam/$\beta$-lactamase system in our study. There are hundreds of other possible drug-target environments to examine. Thankfully, the arithmetic behind the proposed metrics (and the computation of the environmental epistasis that underlines them) is simple, and outlined in the Materials and methods section. To demonstrate how this metric could be applied to different drug classes, we have analyzed a data set of 16 alleles of the *P. falciparum* ortholog of dihydrofolate reductase, in the presence of two different antifolate drugs: pyrimethamine and cycloguanil (see supplemental information). This analysis also demonstrates how the metrics can be computed across a broader range of drug dosages. Note that these data have been examined in prior analyses of fitness landscape topography and epitasis (*Ogbunugafor et al., 2016b*; *Ogbunugafor, 2022*; *Diaz-Colunga et al., 2022*).

### Ideas, speculation, and future directions

Returning to a prior point, we must highlight the resonance between the lens offered from this study, and existing perspectives in the virus-antibody realm. In that case, studies have examined 'binding affinity landscapes' to extract metrics that are akin to our drug applicability (*Doud et al., 2017*; *Doud et al., 2018*), and have characterized virus mutations and epistatic interactions *Phillips et al., 2021*; *Phillips et al., 2023* that are analogous to our variant vulnerability measure. In the virus-antibody problem, the goals are somewhat similar: (i) to identify the characteristics of anti-infectious disease agents that make them likely to be effective against a breadth of variants and (ii) to identify the genetic architecture or signature of pathogens that render these agents ineffective. While the biological problems of antimicrobials-microbes and antibody-virus binding are far different and warrant separate discussions, some methods or metrics can be employed across these domains. At the very least, the convergence in perspectives reflects an urgency to develop new evolutionary-inspired strategies to combat pathogen evolution. This is and will remain a frontier of both evolutionary biology and biomedicine for the foreseeable future. Furthermore, the problem of antimicrobial resistance can more directly employ higher-throughput methods and new technologies to elaborate on variant vulnerability and drug applicability metrics.

Future work may seek to elaborate on these and other studies that highlight the role of environments in crafting plasticity and adaptive evolution (*Cvijović et al., 2015*; *Raghavan et al., 2021*; *Kinsler et al., 2020*). Such efforts may include studies of 'fitness seascapes' and evolution in fluctuating environments, as others have examined (*Mustonen and Lässig, 2009*; *Cvijović et al., 2015*; *King et al., 2022*). These efforts should make use of modern modeling approaches which can rigorously simulate fitness landscapes in realistic, population-level contexts (*Cárdenas et al., 2022*).

### Conclusion

In sum, the findings are relevant to the evolution of antimicrobial resistance and, more broadly, evolutionary medicine. For example, our study supports prior efforts that highlight the importance of higher order interactions in the evolution of antimicrobial resistance (*Weinreich et al., 2006*; *Lozovsky et al., 2021*) and the existence of G x G x E interactions ('environmental epistasis') as a meaningful characteristic of fitness landscapes in microbial systems (*Lindsey et al., 2013*; *Ogbunugafor, 2022*). Generally, this study supports recent efforts that employ evolutionary perspectives to understand antimicrobial resistance (*Michel et al., 2008*; *Yeh et al., 2009*; *Rosenbloom et al., 2012*; *Baym et al., 2016*; *Beppler et al., 2016*; *Gjini and Wood, 2021*; *Ardell and Kryazhimskiy, 2021*). Like many others, we believe that we are only at the beginning of a larger, holistic effort to supplement existing metrics and concepts developed within clinical medicine, all toward more effective therapies that can improve outcomes at the bedside.

## Materials and methods

### Notes on the terminology I

The quantities we define as 'variant vulnerability' and 'drug applicability' may be seen as descendants of 'drugability' (or 'druggability' and other related terms), which are sometimes applied to properties of targets (*Halgren, 2009*; *Jubb et al., 2015*) and other times to drug candidates themselves (*Keller et al., 2006*; *Benet et al., 2016*, p.5). These metrics are generally relegated to molecular properties of drugs and targets, and are most often deployed in medicinal chemistry. To avoid ambiguity and in

**Table 3.** Terms used or introduced in this study and their definition.
For some terms, references are provided for clarity.

| Term | Study definition |
| --- | --- |
| Drugability or Druggability | Generally used to describe the ability to treat a drug target with a small molecule-based drug *Halgren, 2009*; *Jubb et al., 2015*. Sometimes refers to the ability of a drug to be used against a single or set of drug targets, and/or used to describe whether a small molecule fits into a compound class that is or can be adapted to have pharmacologic properties useful in the clinic (e.g., solubility) *Keller et al., 2006*; *Benet et al., 2016*. |
| Cross-resistance | Phenomenon whereby pathogens are resistant to all or most antimicrobials belonging to a given class via a single mechanism. |
| Susceptibility | $1 - w$, where $w$ is the mean growth of the allelic variant under drug conditions relative to the mean growth of the wild-type control. |
| Environmental epistasis | A term that comes from a prior study on G x G x E interactions *Lindsey et al., 2013*. Analysis of the various G x G x E interactions that drive the variant vulnerability and drug applicability metrics. |
| Variant vulnerability | The average susceptibility of an allele across drugs in a panel. |
| Drug applicability | The average susceptibility of all variants to an antibiotic. |

Source: The authors.

striving to bridge concepts from chemical biology, pharmacology, and evolutionary systems biology, we have avoided using either term (druggability or drugability) in favor of new terminology. In light of this, we have selected '*variant vulnerability*' to apply to allelic variants and '*drug applicability*' to apply to drugs. We provide study definitions of these and other terms and metrics in *Table 3*. Also note that our measure of variant vulnerability resonates with the concept of cross-resistance, which applies to pathogens that are resistant by a single mechanism to all or most antimicrobials belonging to a class (*Cândido et al., 2019*; *Colclough et al., 2019*). Some studies highlight that cross-resistance can manifest across genetic variants, with mutations conferring resistance to two drugs simultaneously (*Yeh et al., 2006*). While modern studies have added nuance to our understanding of how resistance and cross-resistance can evolve (*Oz et al., 2014*), ambiguity remains surrounding how it is used and, more importantly, how it is measured. The variant vulnerability metric improves upon notions of cross-resistance in two important ways: (i) it incorporates genetic variation into our understanding of how pathogens can be resistant to a panel of drugs, and (ii) it provides a quantitative metric for such cross-resistance (cross-resistance is typically used qualitatively).

## Notes on the terminology II

In addition to the above conversation surrounding the terms 'drugability' and 'druggability', it is also important to briefly clarify the uses of the term 'fitness landscape' vs. 'adaptive landscape'. These terms can generally be used interchangeably and sometimes to describe how evolution moves through protein space, an idea pioneered by *Maynard Smith, 1970*; *Ogbunugafor, 2020*, that has since been invoked to describe aspects of protein evolution and engineering (*Romero and Arnold, 2009*; *Arnold, 2011*; *Arnold, 2019*). We use fitness landscape in this study, partly because we are talking about a phenotype—growth rate—that is a component of bacterial fitness in the presence of antibiotics (there are, of course, other proxies, like competitive fitness). We also note that the data used in this study comes from a prior study that discussed these data (*Mira et al., 2015*). Because of that, we resisted the use of 'protein space' or other analogies that similarly describe data of this sort.

## Note on the study system

This study utilizes the $\beta$-lactam/$\beta$-lactamase target-pairs for purposes of illustration. $\beta$-lactamases are a class of enzymes produced by bacteria that break down the $\beta$-lactam ring of antibiotics, making them ineffective. TEM -type $\beta$-lactamases are the main culprit of $\beta$-lactam resistance in gram-negative microorganisms. TEM was first identified in *Escherichia coli* isolated from a patient named Temoneria in Greece in 1965 (*Steward et al., 2000*). The evolution and subsequent spread of $\beta$-lactamases arose from the widespread clinical use of $\beta$-lactam antibiotics. As the clinical treatment landscape involves using $\beta$-lactam drugs of various kinds (old and new), the population genetic landscape of $\beta$-lactamases is diverse (*Blazquez et al., 2000*). Among clinical populations of gram-negative microorganisms, the TEM-1 allele is the most frequently detected antibacterial resistance variant (*Mroczkowska and*

*Barlow, 2008*). Many of the most common alleles of other TEM genes are mutants of TEM-1, with TEM-50 a derivative of TEM-1 separated by four mutations: Met → Leu-69, Glu → Lys-104, Gly → Ser-238 and Asn → Asp-276 (*Sirot et al., 1997*). These mutations confer structural, and functional differences that meaningfully influence the clinical use of antibiotics (*Pimenta et al., 2014*). In the decades since $\beta$-lactamases were identified for their public health relevance, they have become one of the key protein systems for evolutionary biochemistry and biophysics studies. Pioneering studies have examined concepts like the adaptive trajectory (*Weinreich et al., 2006*), evolvability (*Stiffler et al., 2015*), and biophysical tradeoffs (*Knies et al., 2017*), and have used techniques like deep-mutational scanning to examine the fitness effects of mutations in high-throughput (*Rollins et al., 2019*). While this study focuses on data from the $\beta$ lactam/$\beta$-lactamase interaction, the methods and understandings can broadly apply to other study systems (e.g. dihydrofolate reductase and antifolate drugs).

In the current study, we describe the set of allelic variants using their single amino acid abbreviations: M69L, E104K, G238S, N276D. We can also utilize binary notation describing the mutants (as do many studies of low-dimension fitness landscapes), with the TEM-1 locus corresponding to a 0, and a 1 corresponding to a mutation associated with TEM-50. For example, the TEM-1 allele, MEGH, can also be described as 0000, and the quadruple mutant as LKSD or 1111.

## Data

The data we examine comes from a past study of a combinatorial set of four mutations associated with TEM-50 resistance to $\beta$-lactam drugs (*Mira et al., 2015*). This past study measured the growth rates of these four mutations in combination, across 15 different drugs (see supplemental information). We examined these data, identifying a subset of structurally similar $\beta$-lactams that also included $\beta$-lactams combined with $\beta$-lactamase inhibitors, cephalosporins and penicillins.

From the original data set (see *Mira et al., 2015* and the supplemental information), we focus our analyses on drug treatments that had a significant negative effect on the growth of wild-type/TEM-1 strains (one-tailed t-test of wild-type treatment vs. control, p<0.01). After identifying the data from the set that fit our criteria, we were left with seven drugs or combinations: amoxicillin 1024 μg/ml ($\beta$-lactam), amoxicillin/clavulanic acid 1024 μg/ml ($\beta$-lactam and $\beta$-lactamase inhibitor) cefotaxime 0.123 μg/ml (third-generation cephalosporin), cefotetan 0.125 μg/ml (second-generation cephalosporins), cefprozil 128 μg/ml (second-generation cephalosporin), ceftazidime 0.125 μg/ml (third-generation cephalosporin), piperacillin, and tazobactam 512/8 μg/ml (penicillin and $\beta$-lactamase inhibitor). With these drugs/mixtures, we were able to embody chemical diversity in the panel. While drug combinations are composed of mixtures, this study did not measure all combinations in their components, but analyzed results as measured in *Mira et al., 2015*.

Lastly, we studied a single system so that we could focus our analyses and have a substantive discussion about the biological details of the protein-drug interactions. However, the broader concepts and metrics offered in this study can be applied more broadly. In the supplemental material, we have analyzed a different dataset, corresponding to a different target-drug: the *Plasmodium falciparum* ortholog of dihydrofolate reductase (DHFR), and various concentrations of two antifolate drugs, pyrimethamine and cycloguanil. These data offer a few additional benefits: while this set examines fewer drugs in number, they do contain a number of drug concentrations, and so their analyses demonstrates how one can examine the problem across different drug dosages.

## Metric calculations

To estimate the two metrics we are interested in, we must first quantify the susceptibility of an allelic variant to a drug. We define susceptibility as $1 - w$, where w is the mean growth of the allelic variant under drug conditions relative to the mean growth of the wild-type/TEM-1 control. If a variant is not significantly affected by a drug (i.e. growth under the drug is not statistically lower than the growth of wild-type/TEM-1 control, by t-test *P*-value <0.01), its susceptibility is zero. Values in these metrics are summaries of susceptibility: the variant vulnerability of an allelic variant is its average susceptibility across drugs in a panel, and the drug applicability of an antibiotic is the average susceptibility of all variants to it.

We further explored the interactions across this fitness landscape and panels of drugs in two additional ways. First, we calculated the variant vulnerability for one-step neighbors, which is the mean variant vulnerability of all alleles one mutational step away from a focal variant. This metric explains

how the variant vulnerability values are distributed across a fitness landscape. Second, we estimated statistical interaction effects on bacterial growth through LASSO regression (implemented in the 'glmnet' R package; *Friedman et al., 2010*). For each drug, we fit a model of relative growth as a function of M69L x E104K x G238S x N276D (i.e. including all interaction terms between the four amino acid substitutions). The effect sizes of the interaction terms from this regularized regression analysis allow us to infer higher order dynamics for susceptibility. We label this calculation as an analysis of 'environmental epistasis'.

### Note on methods used to measure epistasis

Here, we will briefly summarize methods used to study epistasis on fitness landscapes. Several studies of combinatorially complete fitness landscapes use some variation of the Fourier Transform or Taylor formulation. One in particular, the Walsh-Hadamard Transform has been used to measure epistasis across many study systems (*Weinreich et al., 2013*; *Weinreich et al., 2018*; *Ogbunugafor, 2022*). Furthermore, studies have reconciled these methods with others, or expanded upon the Walsh-Hadamard Transform in ways that can accommodate incomplete data sets (*Faure et al., 2023*; *Doro and Herman, 2022*).

The method that we have utilized, the LASSO regression, determines effect sizes for all interactions (alleles and drug environments). It has been utilized for data sets of similar size and structure, on alleles resistant to trimethoprim, an antifolate antibiotic (*Guerrero et al., 2019*). Among many benefits, the method can accommodate gaps in data and responsibly incorporates experimental noise into the calculation.

## Acknowledgements

The authors acknowledge seminar invitations from the Massachusetts Institute of Technology, University of California, San Diego, Brown University, and the Innovative Genomics Institute (University of California, Berkeley), where iterations of the ideas in this manuscript were discussed. The authors acknowledge support from the National Institutes of Health grants R35GM136- –354 (MDS and RMH) and R01AI168166 (MDS and CBO), R35GM147107 (RFG), and the National Science Foundation's Division of Environmental Biology Award Number 2142720 (CBO). The authors also thank the Martin Luther King Jr Visiting Professors and Scholars Program at the Massachusetts Institute of Technology for support (CBO). The authors thank the organizers and participants in the 2022 workshop entitled "Reimagining the Central Dogma" at The Foundations Institute, University of California, Santa Barbara, where ideas related to those covered in this manuscript were discussed. Lastly, the authors thank K Kabengele and S Scarpino for their helpful feedback on the manuscript.

## Additional information

### Competing interests

C Brandon Ogbunugafor: Reviewing editor, *eLife*. The other authors declare that no competing interests exist.

### Funding

| Funder | Grant reference number | Author |
| --- | --- | --- |
| National Institutes of Health | R35GM136--354 | Ra'Mal M Harris<br>Matthew D Shoulders |
| National Institutes of Health | R01AI168166 | Matthew D Shoulders<br>C Brandon Ogbunugafor |
| National Institutes of Health | R35GM147107 | Rafael F Guerrero |
| National Science Foundation | 2142720 | C Brandon Ogbunugafor |

| Funder | Grant reference number | Author |
|---|---|---|

The funders had no role in study design, data collection and interpretation, or the decision to submit the work for publication.

## Author contributions
Rafael F Guerrero, Conceptualization, Data curation, Software, Formal analysis, Validation, Investigation, Visualization, Methodology, Writing - original draft, Writing - review and editing; Tandin Dorji, Data curation, Formal analysis, Writing - review and editing; Ra'Mal M Harris, Investigation, Writing - review and editing; Matthew D Shoulders, Supervision, Investigation, Writing - original draft, Project administration, Writing - review and editing; C Brandon Ogbunugafor, Conceptualization, Data curation, Formal analysis, Supervision, Funding acquisition, Validation, Investigation, Visualization, Methodology, Writing - original draft, Project administration, Writing - review and editing

## Author ORCIDs
Rafael F Guerrero ⓘ http://orcid.org/0000-0002-8451-3609
Ra'Mal M Harris ⓘ http://orcid.org/0000-0002-9537-371X
Matthew D Shoulders ⓘ http://orcid.org/0000-0002-6511-3431
C Brandon Ogbunugafor ⓘ http://orcid.org/0000-0002-1581-8345

Reviewer #1 (Public Review): https://doi.org/10.7554/eLife.88480.3.sa1
Reviewer #2 (Public Review): https://doi.org/10.7554/eLife.88480.3.sa2
Reviewer #3 (Public Review): https://doi.org/10.7554/eLife.88480.3.sa3
Author response https://doi.org/10.7554/eLife.88480.3.sa4

## Additional files

### Supplementary files
• MDAR checklist

### Data availability
Code and data can be found on GitHub: https://github.com/OgPlexus/evodruggability (copy archived at *Ogbunu, 2024*).

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

# Appendix 1

## Supplemental Material

In *Figure 1* (main text), we offer graphs of fitness landscapes corresponding to the 16 TEM mutants of β lactamase, across 7 different drug environments. Here, we offer the rank order changes of alleles for the 7 different fitness landscapes. These rank orders further highlight the dynamics of the topography.

## Relative growth –– ranked within drug

| | Amoxicillin | Amoxicillin/Clavulanic~Acid | Cefotaxime | Cefotetan | Cefprozil | Ceftazidime | Piperacillin/Tazobactam |
|---|---|---|---|---|---|---|---|
| LKSD | 10 | 13 | 11 | 5 | 10 | 11 | 4 |
| LESD | 3 | 5 | 16 | 13 | 6 | 12 | 6 |
| LKGD | 6 | 15 | 3 | 6 | 13 | 10 | 16 |
| LKSN | 2 | 4 | 7 | 9 | 1 | 6 | 5 |
| MKSD | 1 | 9 | 4 | 7 | 2 | 2 | 2 |
| LKGN | 16 | 9 | 8 | 4 | 5 | 7 | 13 |
| LESN | 11 | 7 | 12 | 10 | 8 | 5 | 3 |
| MKSN | 9 | 12 | 13 | 16 | 16 | 15 | 9 |
| LEGD | 12 | 16 | 5 | 8 | 4 | 3 | 11 |
| MKGD | 4 | 3 | 10 | 1 | 9 | 13 | 15 |
| MESD | 7 | 1 | 14 | 12 | 12 | 14 | 1 |
| LEGN | 5 | 14 | 9 | 2 | 14 | 4 | 12 |
| MKGN | 14 | 8 | 6 | 14 | 15 | 1 | 14 |
| MESN | 13 | 5 | 15 | 15 | 7 | 16 | 8 |
| MEGD | 8 | 2 | 1 | 3 | 3 | 9 | 7 |
| MEGN | 15 | 11 | 2 | 11 | 11 | 8 | 10 |

**Appendix 1—figure 1.** Rank orders for the alleles in the fitness landscapes outlined in *Figure 1*.

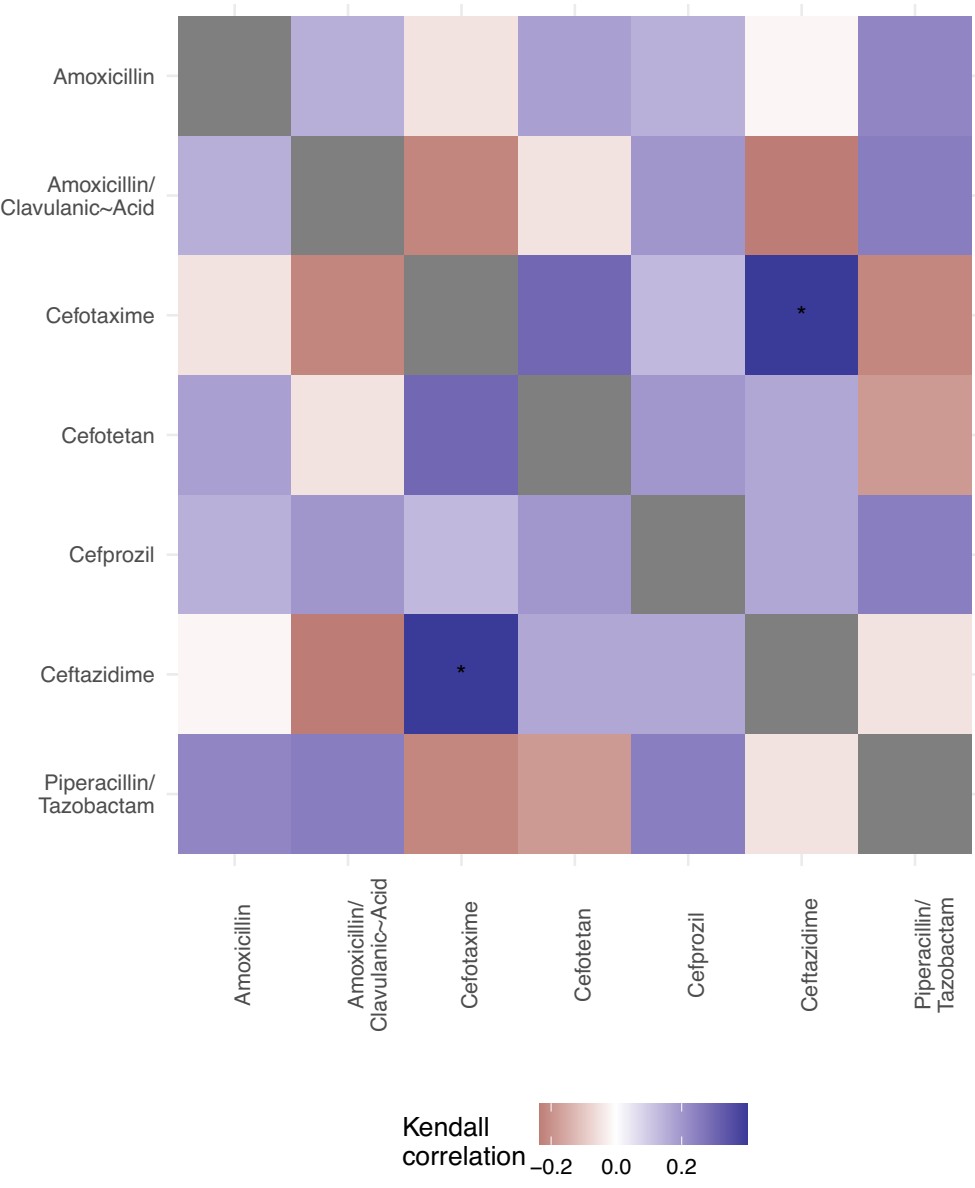

**Appendix 1—figure 2.** We offer an analysis of the topography of those landscapes, based on the Kendall rank order test. This texts the hypothesis that there is no correlation (concordance or discordance) between the topographies of the fitness landscapes.

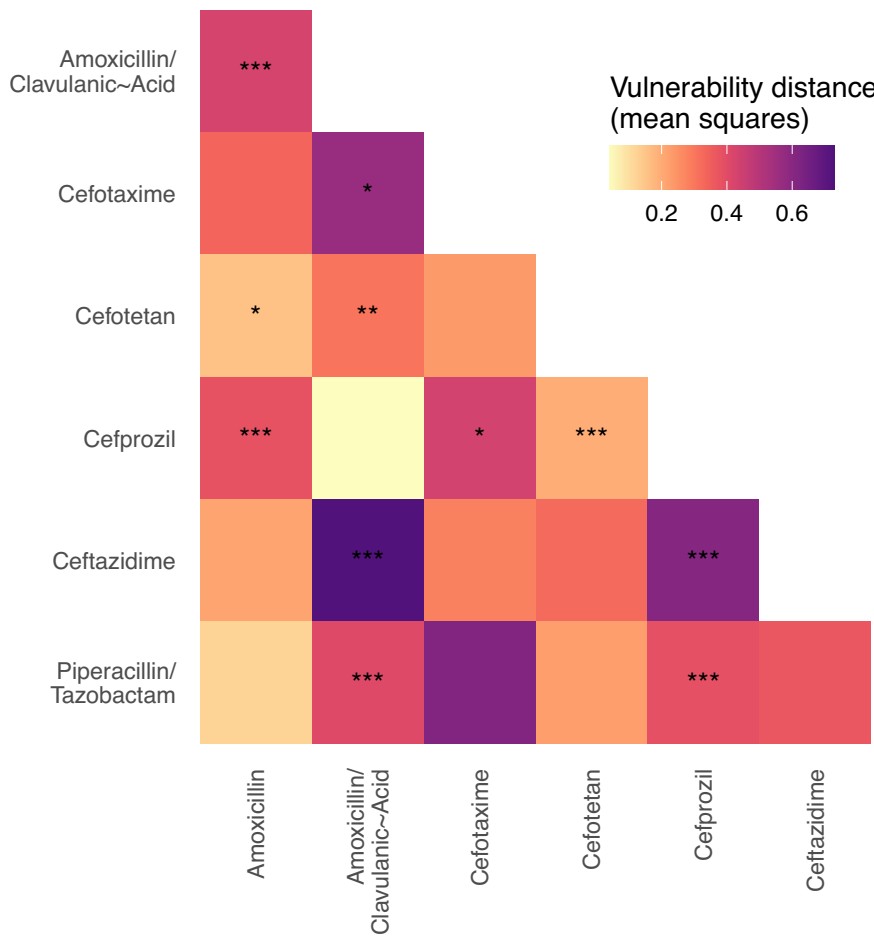

Paired t–test:
\* p < 0.05
\*\* p < 0.01
\*\*\* p < 0.001

**Appendix 1—figure 3.** To test the hypothesis that the variant vulnerability values differ, we calculated a paired t-test. These are paired by haplotype/allelic variant, so the comparisons are change in growth between drugs for each haplotype.

While the study focused on the β lactamase–β lactam system (widely adopted in examinations of the evolution of drug resistance), the metrics are presented such that they might be applied broadly. To demonstrate another use case, we offer a set of 16 alleles of the *Plasmodium falciparum* ortholog of dihydrofolate reductase. These constitute a combinatorial set of mutations, with growth rate across a breadth of concentrations of two antifolate drugs: pyrimethamine (PYR), and cycloguanil (CYC).

In *Appendix 1—figure 4* below, find a figure where we've computed both the variant variability (y-axis) and the drug applicability (x-axis) across the 16 alleles and across 9 concentrations of 2 drugs. Importantly, this analysis demonstrates how the drug applicability metric can be computed across drug concentrations. Previous work has demonstrated that drug concentration can be a meaningful influence on the dynamics of drug resistance in antifolates *Ogbunugafor et al., 2016b*.

As we observe, drug concentration does have a meaningful impact on the metrics. In the main text, we focused on single high doses of each of the 7 drug/mixtures. But the metrics can just as readily be computed across different drug concentrations.

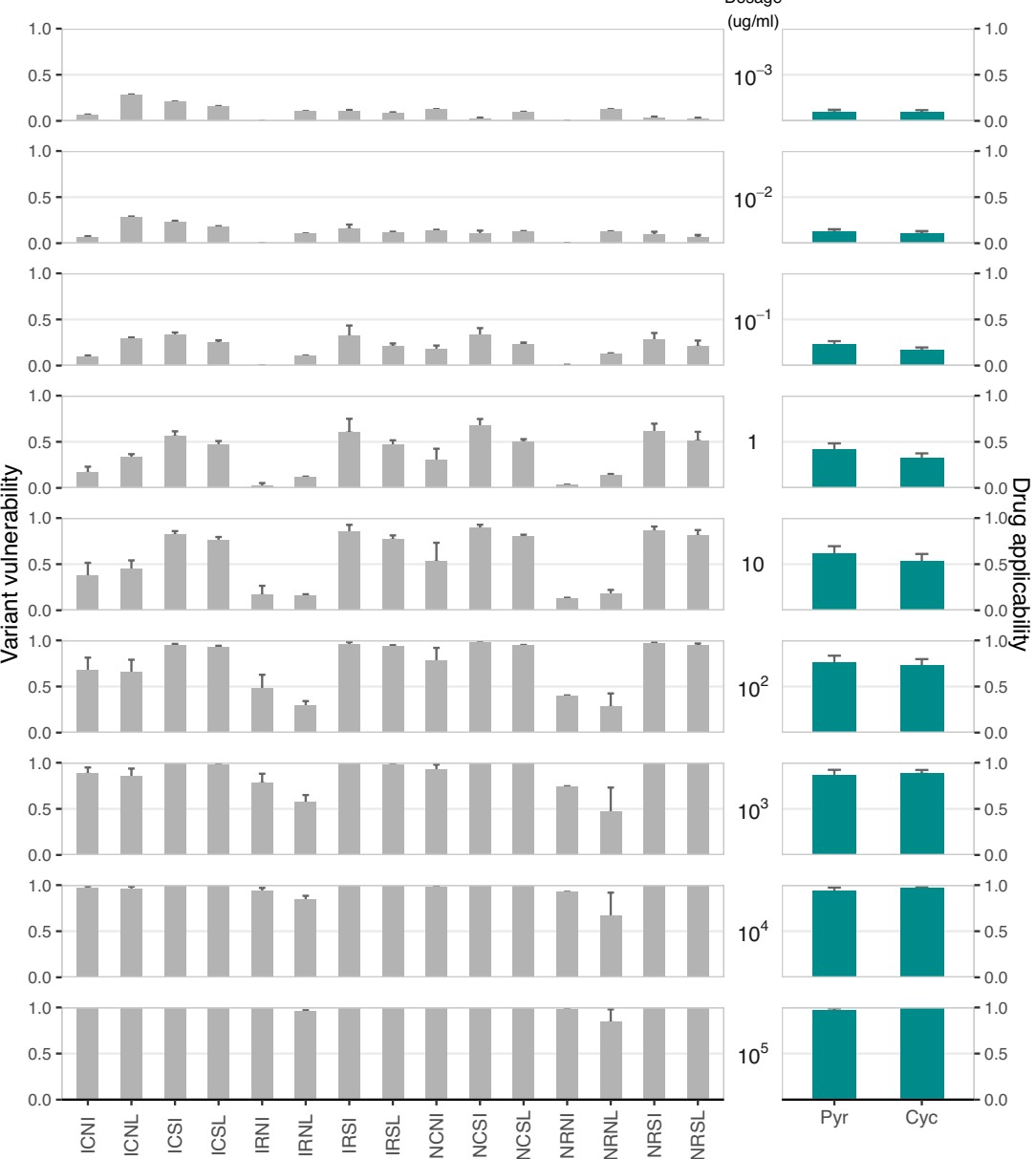

**Appendix 1—figure 4.** Variant vulnerability and drug applicability values for 16 allelic variants of dihydrofolate reductase (*Plasmodium falciparum*), across nine concentrations of two different antifolate drugs.

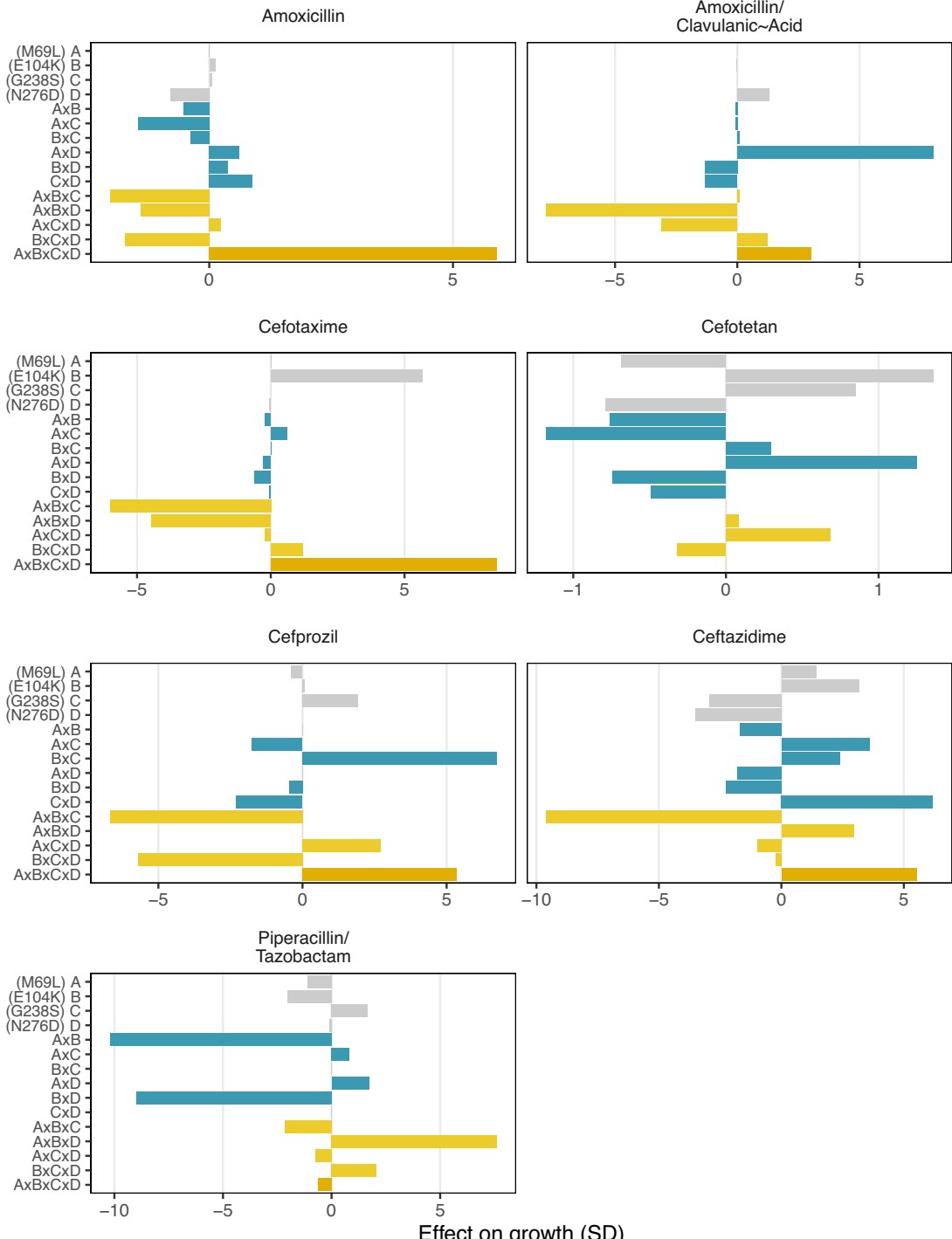

**Appendix 1—figure 5.** As stated in them main text: "Environmental epistasis underlies the drug-allele interactions that drive the variant vulnerability and drug applicability". Effect on growth (in standard deviations of the wild-type control values), estimated by LASSO regression, for individual loci and their interactions. [A] corresponds to M69L, [B] to E104K, [C] to G238S, and [D] to N276D.

