## [Editor Report · eLife assessment]

This manuscript introduces two **valuable** new metrics - "variant vulnerability" and "drug applicability" - that would be of use to identify candidate drugs for treating infections while considering longer-term, evolution-based treatment outcomes. Despite the intuitive appeal of the metrics and their potential, the study remains **incomplete**, as it fails to demonstrate the generality of the approach. The work could be improved by analysing a broader range of data in a systematic way and directly tying the metrics to outcomes, which would make it possible to better assess their impact and utility.

---

## [Referee Report · Reviewer #1 (Public Review)]

The manuscript by Geurrero and colleagues introduces two new metrics that extend the concept of "druggability"- loosely speaking, the potential suitability of a particular drug, target, or drug-target interaction for pharmacological intervention-to collections of drugs and genetic variants. The study draws on previously measured growth rates across a combinatoriality complete mutational landscape involving 4 variants of the TEM-50 (beta lactamase) enzyme, which confers resistance to commonly used beta-lactam antibiotics. To quantify how growth rate - in this case, a proxy for evolutionary fitness - is distributed across allelic variants and drugs, they introduce two concepts: "variant vulnerability" and "drug applicability".

Variant vulnerability is the mean vulnerability (1-normalized growth rate) of a particular variant to a library of drugs, while drug applicability measures the mean across the collection of genetic variants for a given drug. The authors rank the drugs and variants according to these metrics. They show that the variant vulnerability of a particular mutant is uncorrelated with the vulnerability of its one-step neighbors, and analyze how higher-order combinations of single variants (SNPs) contribute to changes in growth rate in different drug environments.

The work addresses an interesting topic and underscores the need for evolution-based metrics to identify candidate pharmacological interventions for treating infections. The authors are clear about the limitations of their approach - they are not looking for immediate clinical applicability - and provide simple new measures of druggability that incorporate an evolutionary perspective, an important complement to the orthodoxy of aggressive, kill-now design principles.

As I said in my initial review, I think the work could be improved with additional analysis that tie the new metrics to evolutionary outcomes. Without this evidence, or some other type of empirical or theoretical support for the utility of these metrics, I am not fully convinced that these concepts have substantial impact. The new metrics could indeed be useful--and they have intuitive appeal--but the current revisions stop short of demonstrating that these intuitive notions hold up under "realistic" conditions (whether in simulation, theory, or experiment).

---

## [Referee Report · Reviewer #2 (Public Review)]

In the main text, the authors apply their metrics to a data set that was published by Mira et al. in 2015. The data consist of growth rate measurements for a combinatorially complete set of 16 genetic variants of the antibiotic resistance enzyme beta-lactamase across 10 drugs and drug combinations at 3 different drug concentrations, comprising a total of 30 different environmental conditions. In my previous report I had asked the authors to specify why they selected only 7 out of 30 environments for their analysis, with only one concentration for drug, but a clear explanation is still lacking. In the Data section of Material and Methods, the authors describe their criterion for data selection as follows: "we focus our analyses on drug treatments that had a significant negative effect on the growth of wildtype/TEM-1 strains". However, in Figure 2 it is seen that, even for the selected data sets, not all points are significant compared to wild type (grey points). So what criterion was actually applied?

In effect, for each chosen drug or drug combination, the authors choose the data set corresponding to the highest drug concentration. As a consequence, they cannot assess to what extent their metrics depend on drug concentration. This is a major concern, since Mira et al. concluded in their study that the differences between growth rate landscapes measured at different concentrations were comparable to the differences between drugs. I argued before that, if the new metrics display a significant dependence on drug concentration, this would considerably limit their usefulness. The authors challenge this, saying in their rebuttal that "no, that drug concentration would

be a major actor in the value of the metrics does not limit the utility of the metric. It is simply another variable that one can consider when computing the metrics." While this is true in principle, I don't think any practicing scientist would disagree with the statement that the existence of additional confounding factors (in particular if they are unknown) reduces the usefulness

of a quantitative metric.

As a consequence of the small number of variant-drug-combinations that are used, the conclusions that the authors draw from their analysis are mostly tentative. For example, on line 123 the authors write that the observation that

the treatment of highest drug applicability is a combination of two drugs "fits intuition". In the Discussion this statement is partly retracted with reference to the piperacillin/tazobactam-combination which has low drug applicability. Being based on only a handful of data points, both observations are essentially anecdotal and it is unclear what the reader is supposed to learn.

To assess the environment-dependent epistasis among the genetic mutations comprising the variants under study, the authors decompose the data of Mira et al. into epistatic interactions of different orders. This part of the analysis is incomplete in two ways. First, in their study, Mira et al. pointed out that a fairly large fraction of the fitness differences between variants that they measured were not statistically significant. This information has been removed in the depiction of the Mira et al. fitness landscapes in Figure 1 of the present manuscript, and it does not seem to be reflected in the results of the interaction analysis in Figure 4. Second, the interpretation of the coefficients obtained from the epistatic decomposition depends strongly on the formalism that is being used. In a note added on page 15 of the revised manuscript, the authors write that they have used the LASSO regression for their analysis and refer the reader to a previous publication (Guerrero et al. 2019) which however (as far as I could see) also does not fully explain how the method works. To give an example of the difficulty of interpreting the data in Figure 4 without further information: The substitution C (G238S) is well known to have a strong positive effective in cefotaxime, but the corresponding coefficient is essentially zero. So whatever the LASSO regression does, it cannot simply measure the effect on growth.

---

## [Referee Report · Reviewer #3 (Public Review)]

The authors introduce two new concepts for antimicrobial resistance borrowed from pharmacology, "variant vulnerability" (how susceptible a particular resistance gene variant is across a class of drugs) and "drug applicability" (how useful a particular drug is against multiple allelic variants). They group both terms under an umbrella term "drugability". They demonstrate these features for an important class of antibiotics, the beta-lactams, and allelic variants of TEM-1 beta-lactamase. In the revised version, they investigate a second drug class that targets dihydrofolate reductase in Plasmodium (the causative agent of malaria).

The strength of the result is in its conceptual advance and that the concepts seem to work for beta-lactam resistance and DHFR inhibitors in a protozoan. However, I do not necessarily see the advance of lumping both terms under "drugability", as this adds an extra layer of complicaton in my opinion.

I think that the utility of the terms will be more comprehensively demonstrated by using examples across a breadth of drug classes classes and/or resistance genes. For instance, another good bacterial model with published data might have been trimethoprim resistance, which arises through point mutations in the folA gene (although, clinical resistance tends to be instead conferred by a suite of horizontally acquired dihydrofolate reductase genes, which are not so closely related as the TEM variants explored here).

The impact of the work on the field depends on a more comprehensive demonstration of the applicability of these new concepts to other drugs. This would be demonstrated in future work.

---

## [Author Response]

The following is the authors’ response to the original reviews.

Firstly, we must take a moment to express our sincere gratitude to editorial board for allowing this work to be reviewed, and to the peer reviewers for taking the time and effort to review our manuscript. The reviews are thoughtful and reflect the careful work of scientists who undoubtedly have many things on their schedule. We cannot express our gratitude enough. This is not a minor sentiment. We appreciate the engagement.

Allow us to briefly highlight some of the changes made to the revised manuscript, most on behalf of suggestions made by the reviewers:

1. A supplementary figure that includes the calculation of drug applicability and variant vulnerability for a different data set–16 alleles of dihydrofolate reductase, and two antifolate compounds used to treat malaria–pyrimethamine and cycloguanil.

2. New supplementary figures that add depth to the result in Figure 1 (the fitness graphs): we demonstrate how the rank order of alleles changes across drug environments and offer a statistical comparison of the equivalence of these fitness landscapes.

3. A new subsection that explains our specific method used to measure epistasis.

4. Improved main text with clarifications, fixed errors, and other addendums.

5. Improved referencing and citations, in the spirit of better scholarship (now with over 70 references).

Next, we’ll offer some general comments that we believe apply to several of the reviews, and to the eLife assessment. We have provided the bulk of the responses in some general comments, and in response to the public reviews. We have also included the suggestions and made brief comments to some of the individual recommendations.

On the completeness of our analysis

In our response, we’ll address the completeness issue first, as iterations of it appear in several of the reviews, and it seems to be one of the most substantive philosophical critiques of the work (there are virtually no technical corrections, outside of a formatting and grammar fixes, which we are grateful to the reviewers for identifying).

To begin our response, we will relay that we have now included an analysis of a data set corresponding to mutants of a protein, dihydrofolate reductase (DHFR), from *Plasmodium falciparum* (a main cause of malaria), across two antifolate drugs (pyrimethamine and ycloguanil). We have also decided to include this new analysis in the supplementary material (see Figure S4).

**Author response image 1. sa4fig1:** Drug applicability and variant vulnerability for 16 alleles of dihydrofolate reductase.

Here we compute the variant vulnerability and drug applicability metrics for two drugs, pyrimethamine (PYR) and cycloguanil (CYC), both antifolate drugs used to treat malaria. This is a completely different system than the one that is the focus of the submitted paper, for a different biomedical problem (antimalarial resistance), using different drugs, and targets. Further, the new data provide information on both drugs of different kinds, and drug concentrations (as suggested by Reviewer #1; we’ve also added a note about this in the new supplementary material). Note that these data have already been the subject of detailed analyses of epistatic effects, and so we did not include those here, but we do offer that reference:

● Ogbunugafor CB. The mutation effect reaction norm (mu-rn) highlights environmentally dependent mutation effects and epistatic interactions. Evolution. 2022 Feb 1;76(s1):37-48.

● Diaz-Colunga J, Sanchez A, Ogbunugafor CB. Environmental modulation of global epistasis is governed by effective genetic interactions. bioRxiv. 2022:202211.

Computing our proposed metrics across different drugs is relatively simple, and we could have populated our paper with suites of similar analyses across data sets of various kinds. Such a paper would, in our view, be spread too thin–the evolution of antifolate resistance and/or antimalarial resistance are enormous problems, with large literatures that warrant focused studies. More generally, as the reviewers doubtlessly understand, simply analyzing more data sets does not make a study stronger, especially one like ours, that is using empirical data to both make a theoretical point about alleles and drugs and offer a metric that others can apply to their own data sets.

Our approach focused on a data set that allowed us to discuss the biology of a system: a far stronger paper, a far stronger proof-of-concept for a new metric. We will revisit this discussion about the structure of our study. But before doing so, we will elaborate on why the “more is better” tone of the reviews is misguided.

We also note that study where the data originate (Mira et al. 2015) is focused on a single data set of a single drug-target system. We should also point out that Mira et al. 2015 made a general point about drug concentrations influencing the topography of fitness landscapes, not unlike our general point about metrics used to understand features of alleles and different drugs in antimicrobial systems.

This isn’t meant to serve as a feeble appeal to authority – just because something happened in one setting doesn’t make it right for another. But other than a nebulous appeal to the fact that things have changed in the 8 years since that study was published, it is difficult to argue why one study system was permissible for other work but is somehow “incomplete” in ours. Double standards can be appropriate when they are justified, but in this case, it hasn’t been made clear, and there is no technical basis for it.

Our study does what countless other successful ones do: utilizes a biological system to make a general point about some phenomena in the natural world. In our case, we were focused on the need for more evolution-inspired iterations of widely used concepts like druggability. For example, a recent study of epistasis focused on a single set of alleles, across several drugs, not unlike our study:

● Lozovsky ER, Daniels RF, Heffernan GD, Jacobus DP, Hartl DL. Relevance of higher-order epistasis in drug resistance. Molecular biology and evolution. 2021 Jan;38(1):142-51.

Next, we assert that there is a difference between an eagerness to see a new metric applied to many different data sets (a desire we share, and plan on pursuing in the future), and the notion that an analysis is “incomplete” without it. The latter is a more serious charge and suggests that the researcher-authors neglected to properly construct an argument because of gaps in the data. This charge does not apply to our manuscript, at all. And none of the reviewers effectively argued otherwise.

Our study contains 7 different combinatorially-complete datasets, each composed of 16 alleles (this not including the new analysis of antifolates that now appear in the revision). One can call these datasets “small” or “low-dimensional,” if they choose (we chose to put this front-and-center, in the title). They are, however, both complete and as large or larger than many datasets in similar studies of fitness landscapes:

● Knies JL, Cai F, Weinreich DM. Enzyme efficiency but not thermostability drives cefotaxime resistance evolution in TEM-1 β-lactamase. Molecular biology and evolution. 2017 May 1;34(5):1040-54.

● Lozovsky ER, Daniels RF, Heffernan GD, Jacobus DP, Hartl DL. Relevance of higher-order epistasis in drug resistance. Molecular biology and evolution. 2021 Jan;38(1):142-51.

● Rodrigues JV, Bershtein S, Li A, Lozovsky ER, Hartl DL, Shakhnovich EI. Biophysical principles predict fitness landscapes of drug resistance. Proceedings of the National Academy of Sciences. 2016 Mar 15;113(11):E1470-8.

● Ogbunugafor CB, Eppstein MJ. Competition along trajectories governs adaptation rates towards antimicrobial resistance. Nature ecology & evolution. 2016 Nov 21;1(1):0007.

● Lindsey HA, Gallie J, Taylor S, Kerr B. Evolutionary rescue from extinction is contingent on a lower rate of environmental change. Nature. 2013 Feb 28;494(7438):463-7.

These are only five of very many such studies, some of them very well-regarded.

Having now gone on about the point about the data being “incomplete,” we’ll next move to the more tangible comment-criticism about the low-dimensionality of the data set, or the fact that we examined a single drug-drug target system (β lactamases, and β-lactam drugs).

The criticism, as we understand it, is that the authors could have analyzed more data,

This is a common complaint, that “more is better” in biology. While we appreciate the feedback from the reviewers, we notice that no one specified what constitutes the right amount of data. Some pointed to other single data sets, but would analyzing two different sets qualify as enough? Perhaps to person A, but not to persons B - Z. This is a matter of opinion and is not a rigorous comment on the quality of the science (or completeness of the analysis).

● Should we analyze five more drugs of the same target (beta lactamases)? And what bacterial orthologs?

● Should we analyze 5 antifolates for 3 different orthologs of dihydrofolate reductase?

● And in which species or organism type? Bacteria? Parasitic infections?

● And why only infectious disease? Aren’t these concepts also relevant to cancer? (Yes, they are.)

● And what about the number of variants in the aforementioned target? Should one aim for small combinatorially complete sets? Or vaster swaths of sequence space, such as the ones generated by deep mutational scanning and other methods?

I offer these options in part because, for the most part, were not given an objective suggestion for appropriate level of detail. This is because there is no answer to the question of what size of dataset would be most appropriate. Unfortunately, without a technical reason why a data set of unspecified size [X] or [Y] is best, then we are left with a standard “do more work” peer review response, one that the authors are not inclined to engage seriously, because there is no scientific rationale for it.

The most charitable explanation for why more datasets would be better is tied to the abstract notion that seeing a metric measured in different data sets somehow makes it more believable. This, as the reviewers undoubtedly understand, isn’t necessarily true (in fact, many poor studies mask a lack of clarity with lots of data).

To double down on this take, we’ll even argue the opposite: that our focus on a single drug system is a strength of the study.

The focus on a single-drug class allows us to practice the lost art of discussing the peculiar biology of the system that we are examining. Even more, the low dimensionality allows us to discuss–in relative detail–individual mutations and suites of mutations. We do so several times in the manuscript, and even connect our findings to literature that has examined the biophysical consequences of mutations in these very enzymes.

(For example: Knies JL, Cai F, Weinreich DM. Enzyme efficiency but not thermostability drives cefotaxime resistance evolution in TEM-1 β-lactamase. Molecular biology and evolution. 2017 May 1;34(5):1040-54.)

Such detail is only legible in a full-length manuscript because we were able to interrogate a system in good detail. That is, the low-dimensionality (of a complete data set) is a strength, rather than a weakness. This was actually part of the design choice for the study: to offer a new metric with broad application but developed using a system where the particulars could be interrogated and discussed.

Surely the findings that we recover are engineered for broader application. But to suggest that we need to apply them broadly in order to demonstrate their broad impact is somewhat antithetical to both model systems research and to systems biology, both of which have been successful in extracting general principles for singular (often simple) systems and models.

An alternative approach, where the metric was wielded across an unspecified number of datasets would lend to a manuscript that is unfocused, reading like many modern machine learning papers, where the analysis or discussion have little to do with actual biology. We very specifically avoided this sort of study.

To close our comments regarding data: Firstly, we have considered the comments and analyzed a different data set, corresponding to a different drug-target system (antifolate drugs, and DHFR). Moreover, we don’t think more data has anything to do with a better answer or support for our conclusions or any central arguments. Our arguments were developed from the data set that we used but achieve what responsible systems biology does: introduces a framework that one can apply more broadly. And we develop it using a complete, and well-vetted dataset. If the reviewers have a philosophical difference of opinion about this, we respect it, but it has nothing to do with our study being “complete” or not. And it doesn’t speak to the validity of our results.

Related: On the dependence of our metrics on drug-target system

Several comments were made that suggest the relevance of the metric may depend on the drug being used. We disagree with this, and in fact, have argued the opposite: the metrics are specifically useful because they are not encumbered with unnecessary variables. They are the product of rather simple arithmetic that is completely agnostic to biological particulars.

We explain, in the section entitled “Metric Calculations:

“To estimate the two metrics we are interested in, we must first quantify the susceptibility of an allelic variant to a drug. We define susceptibility as $1 - w$, where w is the mean growth of the allelic variant under drug conditions relative to the mean growth of the wild-type/TEM-1 control. If a variant is not significantly affected by a drug (i.e., growth under drug is not statistically lower than growth of wild-type/TEM-1 control, by t-test P-value < 0.01), its susceptibility is zero. Values in these metrics are summaries of susceptibility: the variant vulnerability of an allelic variant is its average susceptibility across drugs in a panel, and the drug applicability of an antibiotic is the average susceptibility of all variants to it.”

That is, these can be animated to compute the variant vulnerability and drug applicability for data sets of various kinds. To demonstrate this (and we thank the reviewers for suggesting it), we have analyzed the antifolate-DHFR data set as outlined above.

Finally, we will make the following light, but somewhat cynical point (that relates to the “more data” more point generally): the wrong metric applied to 100 data sets is little more than 100 wrong analyses. Simply applying the metric to a wide number of datasets has nothing to do with the veracity of the study. Our study, alternatively, chose the opposite approach: used a data set for a focused study where metrics were extracted. We believe this to be a much more rigorous way to introduce new metrics.

On the Relevance of simulations

Somewhat relatedly, the eLife summary and one of the reviewers mentioned the potential benefit of simulations. Reviewer 1 correctly highlights that the authors have a lot of experience in this realm, and so generating simulations would be trivial. For example, the authors have been involved in studies such as these:

● Ogbunugafor CB, Eppstein MJ. Competition along trajectories governs adaptation rates towards antimicrobial resistance. Nature ecology & evolution. 2016 Nov 21;1(1):0007.

● Ogbunugafor CB, Wylie CS, Diakite I, Weinreich DM, Hartl DL. Adaptive landscape by environment interactions dictate evolutionary dynamics in models of drug resistance. PLoS computational biology. 2016 Jan 25;12(1):e1004710.

● Ogbunugafor CB, Hartl D. A pivot mutation impedes reverse evolution across an adaptive landscape for drug resistance in Plasmodium vivax. Malaria Journal. 2016 Dec;15:1-0.

From the above and dozens of other related studies, we’ve learned that simulations are critical for questions about the end results of dynamics across fitness landscapes of varying topography. To simulate across the datasets in the submitted study would be be a small ask. We do not provide this, however, because our study is not about the dynamics of de novo evolution of resistance. In fact, our study focuses on a different problem, no less important for understanding how resistance evolves: determining static properties of alleles and drugs, that provide a picture into their ability to withstand a breadth of drugs in a panel (variant vulnerability), or the ability of a drug in a panel to affect a breadth of drug targets.

The authors speak on this in the Introduction:

“While stepwise, de novo evolution (via mutations and subsequent selection) is a key force in the evolution of antimicrobial resistance, evolution in natural settings often involves other processes, including horizontal gene transfer and selection on standing genetic variation. Consequently, perspectives that consider variation in pathogens (and their drug targets) are important for understanding treatment at the bedside. Recent studies have made important strides in this arena. Some have utilized large data sets and population genetics theory to measure cross-resistance and collateral sensitivity. Fewer studies have made use of evolutionary concepts to establish metrics that apply to the general problem of antimicrobial treatment on standing genetic variation in pathogen populations, or for evaluating the utility of certain drugs’ ability to treat the underlying genetic diversity of pathogens”

That is, the proposed metrics aren’t about the dynamics of stepwise evolution across fitness landscapes, and so, simulating those dynamics don’t offer much for our question. What we have done instead is much more direct and allows the reader to follow a logic: clearly demonstrate the topography differences in Figure 1 (And Supplemental Figure S2 and S3 with rank order changes).

**Author response image 2. sa4fig2:** 

These results tell the reader what they need to know: that the topography of fitness landscapes changes across drug types. Further, we should note that Mira et al. 2015 already told the basic story that one finds different adaptive solutions across different drug environments. (Notably, without computational simulations).

In summary, we attempted to provide a rigorous, clean, and readable study that introduced two new metrics. Appeals to adding extra analysis would be considered if they augmented the study’s goals. We do not believe this to be the case.

Nonetheless, we must reiterate our appreciation for the engagement and suggestions. All were made with great intentions. This is more than one could hope for in a peer review exchange. The authors are truly grateful.

**eLife assessment**
The work introduces two valuable concepts in antimicrobial resistance: "variant vulnerability" and "drug applicability", which can broaden our ways of thinking about microbial infections through evolution-based metrics. The authors present a compelling analysis of a published dataset to illustrate how informative these metrics can be, study is still incomplete, as only a subset of a single dataset on a single class of antibiotics was analyzed. Analyzing more datasets, with other antibiotic classes and resistance mutations, and performing additional theoretical simulations could demonstrate the general applicability of the new concepts.

The authors disagree strongly with the idea that the study is ‘incomplete,” and encourage the editors and reviewers to reconsider this language. Not only are the data combinatorially complete, but they are also larger in size than many similar studies of fitness landscapes. Insofar as no technical justification was offered for this “incomplete” summary, we think it should be removed. Furthermore, we question the utility of “theoretical simulations.” They are rather easy to execute but distract from the central aims of the study: to introduce new metrics, in the vein of other metrics–like druggability, IC50, MIC–that describe properties of drugs or drug targets.

**Public Reviews:**

**Reviewer #1 (Public Review):**
The manuscript by Geurrero and colleagues introduces two new metrics that extend the concept of "druggability"- loosely speaking, the potential suitability of a particular drug, target, or drug-target interaction for pharmacological intervention-to collections of drugs and genetic variants. The study draws on previously measured growth rates across a combinatoriality complete mutational landscape involving 4 variants of the TEM-50 (beta lactamase) enzyme, which confers resistance to commonly used beta-lactam antibiotics. To quantify how growth rate - in this case, a proxy for evolutionary fitness - is distributed across allelic variants and drugs, they introduce two concepts: "variant vulnerability" and "drug applicability".Variant vulnerability is the mean vulnerability (1-normalized growth rate) of a particular variant to a library of drugs, while drug applicability measures the mean across the collection of genetic variants for a given drug. The authors rank the drugs and variants according to these metrics. They show that the variant vulnerability of a particular mutant is uncorrelated with the vulnerability of its one-step neighbors and analyze how higher-order combinations of single variants (SNPs) contribute to changes in growth rate in different drug environments.The work addresses an interesting topic and underscores the need for evolutionbased metrics to identify candidate pharmacological interventions for treating infections. The authors are clear about the limitations of their approach - they are not looking for immediate clinical applicability - and provide simple new measures of druggability that incorporate an evolutionary perspective, an important complement to the orthodoxy of aggressive, kill-now design principles. I think the ideas here will interest a wide range of readers, but I think the work could be improved with additional analysis - perhaps from evolutionary simulations on the measured landscapes - that tie the metrics to evolutionary outcomes.

The authors greatly appreciate these comments, and the proposed suggestions by reviewer 1. We have addressed most of the criticisms and suggestions in our comments above.

**Reviewer #2 (Public Review):**
The authors introduce the notions of "variant vulnerability" and "drug applicability" as metrics quantifying the sensitivity of a given target variant across a panel of drugs and the effectiveness of a drug across variants, respectively. Given a data set comprising a measure of drug effect (such as growth rate suppression) for pairs of variants and drugs, the vulnerability of a variant is obtained by averaging this measure across drugs, whereas the applicability of a drug is obtained by averaging the measure across variants.The authors apply the methodology to a data set that was published by Mira et al.in 2015. The data consist of growth rate measurements for a combinatorially complete set of 16 genetic variants of the antibiotic resistance enzyme betalactamase across 10 drugs and drug combinations at 3 different drug concentrations, comprising a total of 30 different environmental conditions. For reasons that did not become clear to me, the present authors select only 7 out of 30 environments for their analysis. In particular, for each chosen drug or drug combination, they choose the data set corresponding to the highest drug concentration. As a consequence, they cannot assess to what extent their metrics depend on drug concentration. This is a major concern since Mira et al. concluded in their study that the differences between growth rate landscapes measured at different concentrations were comparable to the differences between drugs. If the new metrics display a significant dependence on drug concentration, this would considerably limit their usefulness.

The authors appreciate the point about drug concentration, and it is one that the authors have made in several studies.

The quick answer is that whether the metrics are useful for drug type-concentration A or B will depend on drug type-concentration A or B. If there are notable differences in the topography of the fitness landscape across concentration, then we should expect the metrics to differ. What Reviewer #2 points out as a “major concern,” is in fact a strength of the metrics: it is agnostic with respect to type of drug, type of target, size of dataset, or topography of the fitness landscape. And so, the authors disagree: no, that drug concentration would be a major actor in the value of the metrics does not limit the utility of the metric. It is simply another variable that one can consider when computing the metrics.

As discussed above, we have analyzed data from a different data set, in a different drug-target problem (DHFR and antifolate drugs; see supplemental information). These demonstrate how the metric can be used to compute metrics across different drug concentrations.

As a consequence of the small number of variant-drug combinations that are used, the conclusions that the authors draw from their analysis are mostly tentative with weak statistical support. For example, the authors argue that drug combinations tend to have higher drug applicability than single drugs, because a drug combination ranks highest in their panel of 7. However, the effect profile of the single drug cefprozil is almost indistinguishable from that of the top-ranking combination, and the second drug combination in the data set ranks only 5th out of 7.

We reiterate our appreciation for the engagement. Reviewer #2 generously offers some technical insight on measurements of epistasis, and their opinion on the level of statistical support for our claims. The authors are very happy to engage in a dialogue about these points. We disagree rather strongly, and in addition to the general points raised above (that speak to some of this), will raise several specific rebuttals to the comments from Reviewer #2.

For one, the Reviewer #2 is free to point to what arguments have “weak statistical support.” Having read the review, we aren’t sure what this is referring to. “Weak statistical support” generally applies to findings built from underpowered studies, or designs constructed in manner that yield effect sizes or p-values that give low confidence that a finding is believable (or is replicable). This sort of problem doesn’t apply to our study for various reasons, the least of which being that our findings are strongly supported, based on a vetted data set, in a system that has long been the object of examination in studies of antimicrobial resistance.

For example, we did not argue that magnetic fields alter the topography of fitness landscapes, a claim which must stand up to a certain sort of statistical scrutiny. Alternatively, we examined landscapes where the drug environment differed statistically from the non-drug environment and used them to compute new properties of alleles and drugs.

We can imagine that the reviewer is referring to the low-dimensionality of the fitness landscapes in the study. Again: the features of the dataset are a detail that the authors put into the title of the manuscript. Further, we emphasize that it is not a weakness, but rather, allows the authors to focus, and discuss the specific biology of the system. And we responsibly explain the constraints around our study several times, though none of them have anything to do with “weak statistical support.”

Even though we aren’t clear what “weak statistical support” means as offered by Reviewer 2, the authors have nonetheless decided to provide additional analyses, now appearing in the new supplemental material.

We have included a new Figure S2, where we offer an analysis of the topography of the 7 landscapes, based on the Kendall rank order test. This texts the hypothesis that there is no correlation (concordance or discordance) between the topographies of the fitness landscapes.

**Author response image 3. sa4fig3:** Kendall rank test for correlation between the 7 fitness landscapes.

In Figure S3, we test the hypothesis that the variant vulnerability values differ. To do this, we calculate a paired t-test. These are paired by haplotype/allelic variant, so the comparisons are change in growth between drugs for each haplotype.

**Author response image 4. sa4fig4:** Paired t-tests for variant vulnerability.

To this point raised by Reviewer #2:

“For example, the authors argue that drug combinations tend to have higher drug applicability than single drugs, because a drug combination ranks highest in their panel of 7. However, the effect profile of the single drug cefprozil is almost indistinguishable from that of the top-ranking combination, and the second drug combination in the data set ranks only 5th out of 7.”

Our study does not argue that drug combinations are necessarily correlated with a higher drug applicability. Alternatively, we specifically highlight that one of the combinations does not have a high drug applicability:

“Though all seven drugs/combinations are β-lactams, they have widely varying effects across the 16 alleles. Some of the results are intuitive: for example, the drug regime with the highest drug applicability of the set—amoxicillin/clavulanic acid—is a mixture of a widely used β-lactam (amoxicillin) and a β-lactamase inhibitor (clavulanic acid) (see Table 3). We might expect such a mixture to have a broader effect across a diversity of variants. This high applicability is hardly a rule, however, as another mixture in the set, piperacillin/tazobactam, has a much lower drug applicability (ranking 5th out of the seven drugs in the set) (Table 3).”

In general, we believe that the submitted paper is responsible with regards to how it extrapolates generalities from the results. Further, the manuscript contains a specific section that explains limitations, clearly and transparently (not especially common in science). For that reason, we’d encourage reviewer #2 to reconsider their perspective. We do not believe that our arguments are built on “weak” support at all. And we did not argue anything particular about drug combinations writ large. We did the opposite— discussed the particulars of our results in light of the biology of the system.

Thirdly, to this point:

“To assess the environment-dependent epistasis among the genetic mutations comprising the variants under study, the authors decompose the data of Mira et al. into epistatic interactions of different orders. This part of the analysis is incomplete in two ways. First, in their study, Mira et al. pointed out that a fairly large fraction of the fitness differences between variants that they measured were not statistically significant, which means that the resulting fitness landscapes have large statistical uncertainties. These uncertainties should be reflected in the results of the interaction analysis in Figure 4 of the present manuscript.”

The authors are uncertain with regards to the “uncertainties” being referred to, but we’ll do our best to understand: our study utilized the 7 drug environments from Mira et al. 2015 with statistically significant differences between growth rates with and without drug. And so, this point about how the original set contained statistically insignificant treatments is not relevant here. We explain this in the methods section:

“The data that we examine comes from a past study of a combinatorial set of four mutations associated with TEM-50 resistance to β-lactam drugs [39 ]. This past study measured the growth rates of these four mutations in combination, across 15 different drugs (see Supplemental Information).”

We go on to say the following:

“We examined these data, identifying a subset of structurally similar β-lactams that also included β-lactams combined with β-lactamase inhibitors, cephalosporins and penicillins. From the original data set, we focus our analyses on drug treatments that had a significant negative effect on the growth of wild-type/TEM-1 strains (one-tailed ttest of wild-type treatment vs. control, P < 0.01). After identifying the data from the set that fit our criteria, we were left with seven drugs or combinations (concentration in μg/ml): amoxicillin 1024 μg/ ml (β-lactam), amoxicillin/clavulanic acid 1024 μg/m l (βlactam and β-lactamase inhibitor) cefotaxime 0.123 μg/ml (third-generation cephalosporin), cefotetan 0.125 μg/ml (second-generation cephalosporins), cefprozil 128 μg/ml (second-generation cephalosporin), ceftazidime 0.125 μg/ml (third-generation cephalosporin), piperacillin and tazobactam 512/8 μg/ml (penicillin and β-lactamase inhibitor). With these drugs/mixtures, we were able to embody chemical diversity in the panel.”

Again: The goal of our study was to develop metrics that can be used to analyze features of drugs and targets and disentangle these metrics into effects.

Second, the interpretation of the coefficients obtained from the epistatic decomposition depends strongly on the formalism that is being used (in the jargon of the field, either a Fourier or a Taylor analysis can be applied to fitness landscape data). The authors need to specify which formalism they have employed and phrase their interpretations accordingly.

The authors appreciate this nuance. Certainly, how to measure epistasis is a large topic of its own. But we recognize that we could have addressed this more directly and have added text to this effect.

In response to these comments from Reviewer #2, we have added a new section focused on these points (reference syntax removed here for clarity; please see main text for specifics):

“The study of epistasis, and discussions regarding the means to detect and measure now occupies a large corner of the evolutionary genetics literature. The topic has grown in recent years as methods have been applied to larger genomic data sets, biophysical traits, and the "global" nature of epistatic effects. We urge those interested in more depth treatments of the topic to engage larger summaries of the topic.”

“Here will briefly summarize some methods used to study epistasis on fitness landscapes. Several studies of combinatorially-complete fitness landscapes use some variation of Fourier Transform or Taylor formulation. One in particular, the Walsh-Hadamard Transform has been used to measure epistasis across a wide number of study systems. Furthermore, studies have reconciled these methods with others, or expanded upon the Walsh-Hadamard Transform in a way that can accommodate incomplete data sets. These methods are effective for certain sorts of analyses, and we strongly urge those interested to examine these studies.”

“The method that we've utilized, the LASSO regression, determines effect sizes for all interactions (alleles and drug environments). It has been utilized for data sets of similar size and structure, on alleles resistant to trimethoprim. Among many benefits, the method can accommodate gaps in data and responsibly incorporates experimental noise into the calculation.”

As Reviewer #2 understands, there are many ways to examine epistasis on both high and low-dimensional landscapes. Reviewer #2 correctly offers two sorts of formalisms that allow one to do so. The two offered by Reviewer #2, are not the only means of measuring epistasis in data sets like the one we have offered. But we acknowledge that we could have done a better job outlining this. We thank Reviewer #2 for highlighting this, and believe our revision clarifies this.

**Reviewer #3 (Public Review):**
The authors introduce two new concepts for antimicrobial resistance borrowed from pharmacology, "variant vulnerability" (how susceptible a particular resistance gene variant is across a class of drugs) and "drug applicability" (how useful a particular drug is against multiple allelic variants). They group both terms under an umbrella term "drugability". They demonstrate these features for an important class of antibiotics, the beta-lactams, and allelic variants of TEM-1 beta-lactamase.The strength of the result is in its conceptual advance and that the concepts seem to work for beta-lactam resistance. However, I do not necessarily see the advance of lumping both terms under "drugability", as this adds an extra layer of complication in my opinion.

Firstly, the authors greatly appreciate the comments from Reviewer #3. They are insightful, and prescriptive. And allow us to especially thank reviewer 3 for supplying a commented PDF with some grammatical and phrasing suggestions/edits. This is much appreciated. We have examined all these suggestions and made changes.

In general, we agree with the spirit of many of the comments. In addition to our prior comments on the scope of our data, we’ll communicate a few direct responses to specific points raised.

I also think that the utility of the terms could be more comprehensively demonstrated by using examples across different antibiotic classes and/or resistance genes. For instance, another good model with published data might have been trimethoprim resistance, which arises through point mutations in the folA gene (although, clinical resistance tends to be instead conferred by a suite of horizontally acquired dihydrofolate reductase genes, which are not so closely related as the TEM variants explored here).

1. In our new supplemental material, we now feature an analysis of antifolate drugs, pyrimethamine and cycloguanil. We have discussed this in detail above and thank the reviewer for the suggestion.

2. Secondly, we agree that the study will have a larger impact when the metrics are applied more broadly. This is an active area of investigation, and our hope is that others apply our metrics more broadly. But as we discussed, such a desire is not a technical criticism of our own study. We stand behind the rigor and insight offered by our study.

The impact of the work on the field depends on a more comprehensive demonstration of the applicability of these new concepts to other drugs.

The authors don’t disagree with this point, which applies to virtually every potentially influential study. The importance of a single study can generally only be measured by its downstream application. But this hardly qualifies as a technical critique of our study and does not apply to our study alone. Nor does it speak to the validity of our results. The authors share this interest in applying the metric more broadly.

**Reviewer #1 (Recommendations For The Authors):**
The main weakness of the work, in my view, is that it does not directly tie these new metrics to a quantitative measure of "performance". The metrics have intuitive appeal, and I think it is likely that they could help guide treatment options-for example, drugs with high applicability could prove more useful under particular conditions. But as the authors note, the landscape is rugged and intuitive notions of evolutionary behavior can sometimes fail. I think the paper would be much improved if the authors could evaluate their new metrics using some type of quantitative evolutionary model. For example, perhaps the authors could simulate evolutionary dynamics on these landscapes in the presence of different drugs. Is the mean fitness achieved in the simulations correlated with, for example, the drug applicability when looking across an ensemble of simulations with the same drug but varied initial conditions that start from each individual variant? Similarly, if you consider an ensemble of simulations where each member starts from the same variant but uses a different drug, is the average fitness gain captured in some way by the variant vulnerability? All simulations will have limitations, of course, but given that the landscape is fully known I think these questions could be answered under some conditions (e.g. strong selection weak mutation limit, where the model could be formulated as a Markov Chain; see 10.1371/journal.pcbi.1004493 or doi: 10.1111/evo.14121 for examples). And given the authors' expertise in evolutionary dynamics, I think it could be achieved in a reasonable time. With that said, I want to acknowledge that with any new "metrics", it can be tempting to think that "we need to understand it all" before it is useful, and I don't want to fall into that trap here.

The authors respect and appreciate these thoughtful comments.

As Reviewer #1 highlighted, the authors are experienced with building simulations of evolution. For reasons we have outlined above, we don’t believe they would add to the arc of the current story and may encumber the story with unnecessary distractions. Simulations of evolution can be enormously useful for studies focused on particulars of the dynamics of evolution. This submitted study is not one of those. It is charged with identifying features of alleles and drugs that capture an allele’s vulnerability to treatment (variant vulnerability) and a drug’s effectiveness across alleles (drug applicability). Both features integrate aspects of variation (genetic and environmental), and as such, are improvements over both metrics used to describe drug targets and drugs.

The new metrics rely on means, which is a natural choice. Have the authors considered how variance (or other higher moments) might also impact evolutionary dynamics? I would imagine, for example, that the ultimate outcome of a treatment might depend heavily on the shape of the distribution, not merely its mean. This is also something one might be able to get a handle on with simulations.

These are relevant points, and the authors appreciate them. Certainly, moments other than the mean might have utility. This is the reason that we computed the one-step neighborhood variant vulnerability–to see if the variant vulnerability of an allele was related to properties of its mutational neighborhood. We found no such correlation. There are many other sorts of properties that one might examine (e.g., shape of the distribution, properties of mutational network, variance, fano factor, etc). As we don’t have an informed reason to pursue any of this in lieu of others, we are pleased to investigate this in the future.

Also, while we’ve addressed general points about simulations above, we want to note that our analysis of environmental epistasis does consider the variance. We urge Reviewer #1 to see our new section on “Notes on Methods Used to Measure Epistasis” where we explain some of this and supply references to that effect.

As I understand it, the fitness measurements here are measures of per capita growth rate, which is reasonable. However, the authors may wish to briefly comment on the limitations of this choice-i.e. the fact that these are not direct measures of relative fitness values from head-to-head competition between strains.

Reviewer #1 is correct: the metrics are computed from means. As Reviewer 1 definitely understands, debates over what measurements are proper proxies for fitness go back a long time. We added a slight acknowledgement about the existence of multiple fitness proxies in our revision.

The authors consider one-step variant vulnerability. Have the authors considered looking at 2-step, 3-step, etc analogs of the 1-step vulnerability? I wonder if these might suggest potential vulnerability bottlenecks associated with the use of a particular drug/drug combo or trajectories starting from particular variants.

This is an interesting point. We provided one-step values as a means of interrogating the mutational neighborhood of alleles in the fitness landscape. While there could certainly be other pattern-relationships between the variant vulnerability and features of a fitness landscape (as the reviewer recognizes), we don’t have a rigorous reason to test them, other than an appeal to “I would be curious if [Blank].” As in, attempting to saturate the paper with these sorts of examinations might be fun, could turn up an interesting result, but this is true for most studies.

To highlight just how serious we are about future questions along these lines, we’ll offer one specific question about the relationship between metrics and other features of alleles or landscapes. Recent studies have examined the existence of “evolvabilityenhancing mutations,” that propel a population to high-fitness sections of a fitness landscape:

● Wagner, A. Evolvability-enhancing mutations in the fitness landscapes of an RNA and a protein. Nat Commun 14, 3624 (2023). https://doi.org/10.1038/s41467023-39321-8

One present and future area of inquiry involves whether there is any relationship between metrics like variant vulnerability and these sorts of mutations.

We thank Reviewer 1 for engagement on this issue.

Fitness values are measured in the presence of a drug, but it is not immediately clear how the drug concentrations are chosen and, more importantly, how the choice of concentration might impact the landscape. The authors may wish to briefly comment on these effects, particularly in cases where the environment involves combinations of drugs. There will be a "new" fitness landscape for each concentration, but to what extent do the qualitative features changes-or whatever features drive evolutionary dynamics--change?

This is another interesting suggestion. We have analyzed a new data set for dihydrofolate reductase mutants that contains a range of drug concentrations of two different antifolate drugs. The general question of how drug concentrations change evolutionary dynamics has been addressed in prior work of ours:

● Ogbunugafor CB, Wylie CS, Diakite I, Weinreich DM, Hartl DL. Adaptive landscape by environment interactions dictate evolutionary dynamics in models of drug resistance. PLoS computational biology. 2016 Jan 25;12(1):e1004710.

● Ogbunugafor CB, Eppstein MJ. Competition along trajectories governs adaptation rates towards antimicrobial resistance. Nature ecology & evolution. 2016 Nov 21;1(1):0007.

There are a very large number of environment types that might alter the drug availability or variant vulnerability metrics. In our study, we used an established data set composed of different alleles of a Beta lactamase, with growth rates measured across a number of drug environments. These drug environments consisted of individual drugs at certain concentrations, as outlined in Mira et al. 2015. For our study, we examined those drugs that had a significant impact on growth rate.

For a new analysis of antifolate drugs in 16 alleles of dihydrofolate reductase(*Plasmodium falciparum*), we have examined a breadth of drug concentrations (Supplementary Figure S4). This represents a different sort of environment that one can use to measure the two metrics (variant vulnerability or drug applicability). As we suggest in the manuscript, part of the strength of the metric is precisely that it can incorporate drug dimensions of various kinds.

The metrics introduced depend on the ensemble of drugs chosen. To what extent are the chosen drugs representative? Are there cases where nonrepresentative ensembles might be advantageous?

The authors thank the reviewer for this. The general point has been addressed in our comments above. Further, the general question of how a study of one set of drugs applies to other drugs applies to every study of every drug, as no single study interrogates every sort of drug ensemble. That said, we’ve explained the anatomy of our metrics, and have outlined how it can be directly applied to others. There is nothing about the metric itself that has anything to do with a particular drug type – the arithmetic is rather vanilla.

**Reviewer #2 (Recommendations For The Authors):**
1. Regarding my comment about the different formalisms for epistatic decomposition analysis, a key reference isPoelwijk FJ, Krishna V, Ranganathan R (2016). The Context-Dependence of Mutations: A Linkage of Formalisms. PLoS Comput Biol 12(6): e1004771.

The authors appreciate this, are fans of this work, and have cited it in the revision.

An example where both Fourier and Taylor analyses were carried out and the different interpretations of these formalisms were discussed isUnraveling the causes of adaptive benefits of synonymous mutations in TEM-1 βlactamase. Mark P. Zwart, Martijn F. Schenk, Sungmin Hwang, Bertha Koopmanschap, Niek de Lange, Lion van de Pol, Tran Thi Thuy Nga, Ivan G.Szendro, Joachim Krug & J. Arjan G. M. de Visser Heredity 121:406-421 (2018)

The authors are grateful for these references. While we don’t think they are necessary for our new section entitled “Notes on methods used to detect epistasis,” we did engage them, and will keep them in mind for other work that more centrally focuses on methods used to detect epistasis. As the author acknowledges, a full treatment of this topic is too large for a single manuscript, let alone a subsection of one study. We have provided a discussion of it, and pointed the readers to longer review articles that explore some of these topics in good detail:

● C. Bank, Epistasis and adaptation on fitness landscapes, Annual Review of Ecology, Evolution, and Systematics 53 (1) (2022) 457–479.

● T. B. Sackton, D. L. Hartl, Genotypic context and epistasis in individuals and populations, Cell 166 (2) (2016) 279–287.

● J. Diaz-Colunga, A. Skwara, J. C. C. Vila, D. Bajic, Á. Sánchez, Global epistasis and the emergence of ecological function, BioRxviv

1. Although the authors label Figure 4 with the term "environmental epistasis", as far as I can see it is only a standard epistasis analysis that is carried out separately for each environment. The analysis of environmental epistasis should instead focus on which aspects of these interactions are different or similar in different environments, for example, by looking at the reranking of fitness values under environmental changes [see Ref.[26] as well as more recent related work,e.g. Gorter et al., Genetics 208:307-322 (2018); Das et al., eLife9:e55155 (2020)]. To some extent, such an analysis was already performed by Mira et al., but not on the level of epistatic interaction coefficients.

The authors have provided a new analysis of how fitness value rankings have changed across drug environments, often a signature of epistatic effects across environments (Supplementary Figure S1).

We disagree with the idea that our analysis is not a sort of environmental epistasis; we resolve coefficients between loci across different environments. As with every interrogation of G x E effects (G x G x E in our case), what constitutes an “environment” is a messy conversation. We have chosen the route of explaining very clearly what we mean:

“We further explored the interactions across this fitness landscape and panels of drugs in two additional ways. First, we calculated the variant vulnerability for 1-step neighbors, which is the mean variant vulnerability of all alleles one mutational step away from a focal variant. This metric gives information on how the variant vulnerability values are distributed across a fitness landscape. Second, we estimated statistical interaction effects on bacterial growth through LASSO regression. For each drug, we fit a model of relative growth as a function of M69L x E104K x G238S x N276D (i.e., including all interaction terms between the four amino acid substitutions). The effect sizes of the interaction terms from this regularized regression analysis allow us to infer higher-order dynamics for susceptibility. We label this calculation as an analysis of “environmental epistasis.”

As the grammar for these sorts of analyses continues to evolve, the best one can do is be clear about what they mean. We believe that we communicated this directly and transparently.

1. As a general comment, to strengthen the conclusions of the study, it would be good if the authors could include additional data sets in their analysis.

The authors appreciate this comment and have given this point ample treatment. Further, other main conclusions and discussion points are focused on the biology of the system that we examined. Analyzing other data sets may demonstrate the broader reach of the metrics, but it would not alter the strength of our own conclusions (or if they would, Reviewer #2 has not told us how).

1. There are some typos in the units of drug concentrations in Section 2.4 that should be corrected.

The authors truly appreciate this. It is a great catch. We have fixed this in the revised manuscript.

**Reviewer #3 (Recommendations For The Authors):**
I would suggest demonstrating the concepts for a second drug class, and suggest folA variants and trimethoprim resistance, for which there is existing published data similar to what the authors have used here (e.g. Palmer et al. 2015, https://doi.org/10.1038/ncomms8385)

The authors appreciate this insight. As previously described, we have analyzed a data set of folA mutants for the *Plasmodium falciparum* ortholog of dihydrofolate reductase, and included these results in new supplemental material. Please see the supplementary material.

There are some errors in formatting and presentation that I have annotated in a separate PDF file, as the absence of line numbers makes indicating specific things exceedingly difficult.

The authors apologize for the lack of line numbers (an honest oversight), but moreover, are tremendously grateful for this feedback. We have looked at the suggested changes carefully and have addressed many of them. Thank you.

One thing to note: we have included a version of Figure 4 that has effects on the same axes. It appears in the supplementary material (Figure S4).

In closing, the authors would like to thank the editors and three anonymous reviewers for engagement and for helpful comments. We are confident that the revised manuscript qualifies as a substantive revision, and we are grateful to have had the opportunity to participate.